# Layer 5 myelination gates corticothalamic coincidence detection

Nora Jamann [1,2,6], Jorrit S. Montijn [3], Naomi Petersen[1], Roeland Lokhorst[4], Daan van den Burg[1], Maayke Balemans[1], Stan L. W. Driessens [1,7], J. Alexander Heimel [3,5] & Maarten H. P. Kole [1,2]✉

Myelin is essential for the rapid conduction of action potentials (APs) but its role in long-range processing of disparate inputs remains unclear. Here, using a cell-type-specific approach we recorded optogenetically evoked pyramidal neuron responses via in vivo juxtacellular patch-clamp and Neuropixels probes, tracking spike transmission from layer 5 (L5) to the posteromedial thalamic nucleus (POm) in the mouse. Cuprizone-induced demyelination caused millisecond-scale delays, increased temporal jitter and impaired transmission of high-frequency AP bursts. Computational modeling of the saltatory propagation from L5 to POm revealed that myelin loss from neocortical internodes acts as a low-pass filter, impeding high-frequency spikes within the burst. Finally, pairing optogenetic stimulation with whisker input showed that intact myelination is crucial for coincidence detection in the thalamus. These findings suggest that the continuous myelin pattern of L5 axons not only speeds conduction but also enables precise temporal integration of sensory and cortical signals across long-range pathways.

The myelin sheath is a central evolutionary development in vertebrates, providing metabolic support and changing the action potential (AP) propagation from continuous to rapid saltation, speeding conduction velocities 10- to 100-fold[1-3]. The increase in transmission speed reduces AP delay times and temporal dispersion along axonal tracts, which is thought to play an important role in long-range network synchronization[4,5]. For example, myelin patterns in the cortex differ for individual axon tracts to adjust timing delays between axons of different lengths. Experimental studies in the auditory system and corticothalamic- and thalamocortical pathways show that such adaptive myelination ensures isochronicity of spike arrival times to optimize temporal processing across long-range connections[6-9]. In addition, mounting evidence indicates that neuronal activity can regulate axonal properties such as node of Ranvier (noR) geometry, myelin sheath length and/or thickness, mediating a fine-tuning of temporal delays across active fiber tracts through activity and experience-dependent myelin plasticity[10-12]. Although myelin profoundly shapes the spatiotemporal dynamics of precise timing, its impact on circuit computations between distal brain regions is not well understood.

A prominent myelinated projection from the cortex emerges from layer 5 (L5). L5 pyramidal neurons, which project subcortically, also called "extra-telencephalic (ET)," or "pyramidal tract" neurons[13], present some of the longest axons of the nervous system. Their axons carry high-frequency bursts of APs and travel through the corpus

[1]Department of Axonal Signaling, Netherlands Institute for Neuroscience, Royal Netherlands Academy of Arts and Science, Amsterdam, The Netherlands. [2]Cell Biology, Neurobiology and Biophysics, Department of Biology, Faculty of Science, University of Utrecht, Utrecht, The Netherlands. [3]Department of Circuits, Structure and Function, Netherlands Institute for Neuroscience, Royal Netherlands Academy of Arts and Science, Amsterdam, The Netherlands. [4]Department of Mechatronics, Netherlands Institute for Neuroscience, Royal Netherlands Academy of Arts and Science, Amsterdam, The Netherlands. [5]Department of Neurobiology, Donders Institute for Brain, Cognition and Behaviour and Faculty of Science, Radboud University, Nijmegen, The Netherlands. [6]Present address: Institute of Physiology I, Medical Faculty, University of Freiburg, Freiburg, Germany. [7]Present address: Department of Integrative Neurophysiology, Amsterdam Neuroscience, Center for Neurogenomics and Cognitive Research (CNCR), Vrije Universiteit Amsterdam, Amsterdam, The Netherlands. ✉e-mail: m.kole@nin.knaw.nl

callosum from where multiple and diverging pathways arise, including projections to the thalamus, brainstem, and spinal cord[14,15]. Myelination of this class of L5 axons begins immediately adjacent to the axon initial segment (AIS) and, in comparison to other neurons of the neocortex, is unique in its continuous myelin pattern along the primary axon[16–19]. By reducing the L5 internode capacitance, myelin increases the conduction velocity (CV)[3,19,20], but whether temporal accuracy, controlled by myelination, is important for distal brain regions with which L5 neurons are synaptically connected is not well understood. In rodents, one key corticofugal projection from L5 pyramidal neurons arises from the barrel field in primary somatosensory cortex (S1bf), from which axons en route to the brainstem and spinal cord, branch off a large axon collateral innervating the posteromedial nucleus of the thalamus (POm), a higher-order thalamic nucleus[14,21–23]. The L5–POm synapse is a classical "driver" synapse, characterized by a giant presynaptic terminal located at proximal dendrites with high release-probability[22,24,25]. Simultaneous in vivo recordings from cortex and POm revealed that L5 driver synapses in POm effectively relay whisker information from only a few spiking pyramidal neurons, followed by a powerful synaptic depression, causing a rapid attenuation of the POm excitatory postsynaptic potentials (EPSPs) during cortical up-states. The synaptic depression, however, recovers during periods of presynaptic silence or can be overcome during synchronized inputs at multiple giant L5–POm synapses[24,26,27]. In this scheme, sensory transmission in POm is contingent upon the mode of cortical L5 activity. The relative timing of sensory bottom-up and cortical top-down activity is required to gate whisker input into the thalamus, providing a Boolean AND operation[24,28]. Experiments using in vivo optogenetic activation of L5 neurons synchronized with whisker stimulation showed the temporal window for coincidence detection of activity from the ascending whisker pathway is narrowly tuned between ~5 and 30 ms[26].

Here, we tested the hypothesis that myelination of L5 pyramidal neuron axons is critical for computing coincidence detection in the thalamus. Retrograde viral labeling approaches in combination with optogenetics and in vivo juxtacellular and whole-cell recordings in anesthetized mice enabled studying the role of myelination of L5 pyramidal neuron axons for precise timing of corticothalamic feedback to the L5–POm giant synapses. We find that cuprizone-induced demyelination mostly affects gray matter myelination in the neocortex and partially in the thalamus, without changing white-matter regions of the L5–POm pathway. Juxtacellular and Neuropixels recordings, as well as computational models show that gray matter myelin loss suffices to delay spike arrival times and impede the transfer of the burst response. Finally, whisker stimulation shows that the cortical delay impairs integration of corticothalamic feedback with ascending sensory information, indicating a critical role of continuous myelination of L5 pyramidal neuron axons in the neocortex for long-range feedback processing and gating coincidence detection in the thalamus.

## Results

### Targeted visualization and stimulation of corticothalamic axons
To label L5–POm projections we injected a retrograde adeno-associated virus (AAV) expressing double-floxed humanized channelrhodopsin 2 (ChR2) fused to mCherry under the EF1alpha promoter (pAAV-EF1a-double-floxed-hChR2(H134R)-mCherry-WPRE-HGHpA) into the posteromedial nucleus of the thalamus (POm) of *Rbp4*-Cre mice, a L5 specific Cre mouse line (Fig. 1a). Immunofluorescent staining showed selective and dense expression of ChR2 in L5 neurons sending corticofugal axons traversing via the inner capsula and densely innervating the POm (Fig. 1a, b). To examine target sites beyond the thalamus we developed a sparser labeling of corticothalamic projections using a Cre-dependent Flipase virus (Methods). After clearing with iDISCO+ and lightsheet imaging, we could 3D visualize and reconstruct the axonal pathway, revealing that the projections

continued into the brainstem and distally into the spinal cord, in accordance with previous findings[15] (Fig. 1b, Supplementary Movie 1). With the original Cre-dependent retrograde AAV approach, used for all remaining experiments below, the percentage of ChR2-expressing neurons in L5 was on average $9.22 \pm 2.96\%$ (Fig. 1b, c), consistent with previous reports on the fraction of POm-projecting neurons in L5[29]. Furthermore, mCherry⁺ neurons displayed a characteristic thick-tufted dendritic morphology, in agreement with an ET identity of L5 pyramidal neurons[13,29,30] (Fig. 1b, d). Using in vitro whole-cell voltage recordings in acute preparations of thalamocortical slices, the L5–POm pyramidal neurons displayed either a regular ($n = 4/8$) or burst firing pattern ($n = 4/8$, Fig. 1e). Also, stimulation with blue laser light (470 nm, $n = 4$ PN) resulted in rapid-onset burst firing in vitro (Fig. 1f, g).

To examine the synaptic properties of these L5–POm projections, we performed whole-cell patch-clamp recordings in the POm region of the thalamocortical slice, characterized by dense mCherry⁺ corticothalamic fibers (Fig. 1h). Optogenetic activation of the mCherry⁺ corticothalamic fibers evoked rapid and large postsynaptic responses in POm neurons, which could drive AP bursts, consistent with the low-threshold rebound burst responses of thalamocortical neurons (Fig. 1i, j). In the POm region, giant mCherry⁺ boutons (~$4.17 \pm 2.9 \, \mu m$ in diameter, $n = 291$, $N = 5$ mice) were positive for vesicular glutamate transporter 1 protein (vGlut1, Fig. 1h, k), consistent with driver synapses of the L5–POm pathway[21,22,31,32]. Further in line with giant synapse properties, optogenetically evoked responses showed paired-pulse depression in current- and voltage-clamp mode (Fig. 1l)[22,25]. The postsynaptic depression was significantly stronger at higher frequencies (Fig. 1m).

Finally, to determine the properties of spike transmission in the corticothalamic pathway, we employed in vivo juxtacellular patch-clamp recordings in the S1bf or POm in anesthetized, head-fixed mice in combination with optogenetic stimulation of L5 neurons via a fiber-coupled LED (480 nm), placed above S1bf (Fig. 1n). We optically activated ChR2 with trains of short pulses of blue light (5 pulses of 20 ms duration, 1 Hz, 10 mW) and recorded juxtacellular APs from L5 or POm neurons that responded to light (Fig. 1n). The recording locations of optotagged neurons were post-hoc confirmed by alignment of pipette tracks or juxtasomally filled cells to the Allen brain atlas with the MATLAB toolbox for probe alignment "AP histology" (https://github.com/petersaj/AP_histology, Supplementary Fig. 1a–c). Optogenetic stimulation at 10 mW typically generated a burst of 3–4 APs in L5 neurons and 1–2 APs in POm neurons time locked to the stimulus (Fig. 1o). Together, the findings show that we could selectively study spike transmission along the long-range corticothalamic axonal projection from L5 to POm in vivo.

### Demyelination impairs the precision of long-range spike timing
To disrupt the myelin pattern of L5 axons, we subjected mice to a 6-week treatment with 0.2% cuprizone, well established to robustly affect myelination in L5 and L6[19,33]. Cuprizone-induced demyelination is, however, variable in the white matter regions along the dorsoventral and rostro-caudal axis of the brain[34,35]. We examined which areas were affected along the L5–POm axonal tract by quantifying myelin basic protein (MBP) immunofluorescence between control (Ctrl) and cuprizone-treated (Cpz) mice in three regions; (i) cortex (Ctx), (ii) white matter (WM) and (iii) thalamus (Thal) (Fig. 2a, Supplementary Fig. 2a). The total amount of myelin, as measured by MBP covered volume (% of total volume) in the L5 and 6 regions of the cortex was significantly reduced (Fig. 2b, Supplementary Fig. 2a). In contrast, MBP levels were unchanged within the specific white matter tracts and posterior thalamus (Fig. 2b, Supplementary Fig. 2a).

How does the cortical demyelination affect the speed, fidelity, and precision of AP transmission along L5–POm axons? To answer this question, we repeatedly optogenetically stimulated L5 neurons with an optic fiber that was placed above the S1bf and recorded spike onset

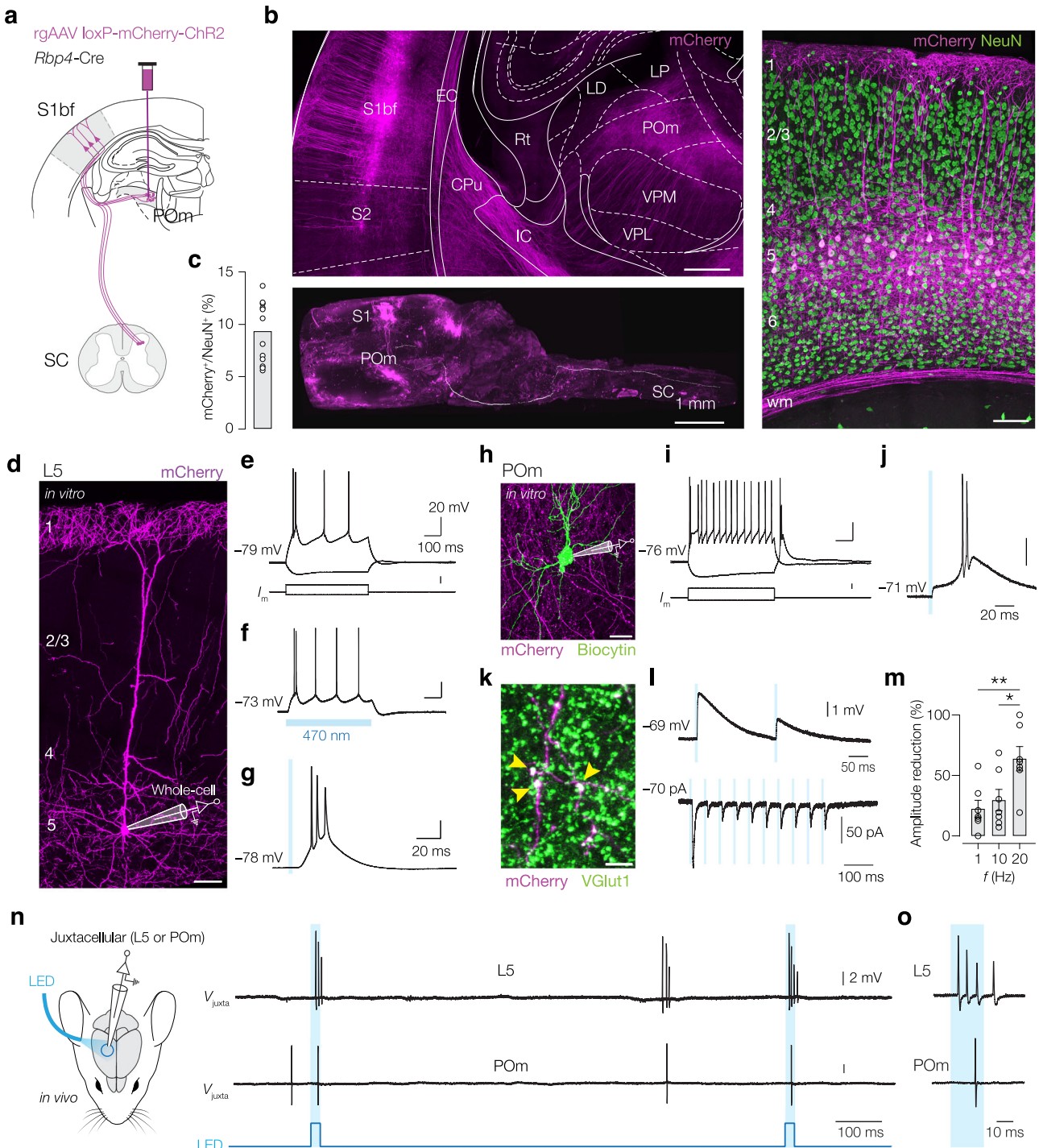

juxtacellularly in the POm. When we measured the delay to spiking (onset light to AP peak), we observed a significant increase in the delay in Cpz mice (Fig. 2c, e, Supplementary Fig. 2b). In addition, spike timing showed more jitter, reflected by an increase in variance (Fig. 2e–g). The observed delay might, besides myelin loss, be explained by a difference in L5 spiking onset, and depend on changes in efficacy of the light stimulation. To test the role of optogenetic activation, we juxtacellularly recorded spiking in L5 and POm for increasing powers (0–25 mW). As expected, at low power (<5 mW) L5 neurons fired APs with variable onset times, in both Ctrl and Cpz mice (Fig. 2f, Supplementary Fig. 2d). However, with increasing power, optogenetic activation of L5 neurons typically generated rapid-onset firing (~4 ms from stimulus onset, at 25 mW) with low onset timing

variance (<1 ms² at 25 mW, Fig. 2f, Supplementary Fig. 2d). Importantly, this power-delay relationship was only slightly affected by demyelination, with Cpz neurons firing significantly earlier at low light power without change in variance (Fig. 2f). The increase of light power also shortened the time from onset to spiking in the POm (Fig. 2g, Supplementary Fig. 2d). However, the delay and variance in spike onset in Cpz mice was independent of the power, indicating that these temporal differences arise after AP initiation, when the spike travels along the L5 axon. When we plotted the delay as a function of POm spike onset relative to L5 onset, control POm neurons fired consistently ~8 ms later. In Cpz mice, however, this delay was significantly increased by ~4 ms (Fig. 2h). In addition, the histogram of all POm spikes showed a lack of fast responses in Cpz neurons (Fig. 2i).

**Fig. 1 | Optotagging and characterization of L5–POm projections. a** Labeling of L5–POm axons with rgAAV expressing floxed-mCherry ChR2 (magenta) injected in POm of *Rbp4*-Cre mice. S1bf, primary somatosensory cortex, barrel field; SC, spinal cord; POm, posteromedial nucleus of the thalamus. **b** Confocal overview image of mCherry axons (*left top*). Lightsheet imaging reveals projections to SC (*left bottom*). mCherry⁺ somata are located in L5, NeuN (green, *right*). Scale bars, 500 μm, 2 mm, and 50 μm. S2 secondary somatosensory cortex, EC external capsule, Cpu caudate putamen, IC internal capsule, Rt reticular nucleus of the thalamus, LD laterodorsal thalamic nucleus, LP lateral posterior thalamic nucleus, VPM ventral posteromedial thalamic nucleus, VPL ventral posterolateral thalamic nucleus, wm white matter. **c** Bar plot of mCherry⁺ NeuN⁺ neurons (2491 NeuN⁺ neurons in total in S1bf L5, *n* = 12 confocal *z*-stacks, *N* = 4 mice. **d** Confocal image of a single mCherry⁺ L5 pyramidal neuron with an illustration of a patch-clamp recording. Scale bar, 50 μm. Recordings replicated in *n* = 8 neurons, *N* = 5 mice. **e** Current clamp recording from a burst-firing mCherry⁺ neuron. Scale bars, 100 ms, 20 mV, and 250 pA. AP burst evoked by a 500 ms and 10 ms blue light stimulation (blue shading) in vitro. Scale bars, 20 mV and 100 ms (**f**), and 20 ms (**g**). **h** Confocal image of a POm neuron (biocytin fill, green) receiving large mCherry⁺ boutons (magenta). Scale bar, 30 μm. In vitro voltage recordings replicated in *n* = 7 POm neurons from *N* = 6 mice. **i** Typical

depolarizing and hyperpolarizing responses from a POm neuron. Scale bars, 20 mV, 100 ms, and 200 pA. **j** Optogenetically evoked AP burst to blue light stimulation (3 ms). Scale bar, 20 mV. **k** mCherry⁺ boutons positive for vesicular glutamate transporter 1 (VGlut1, green). *n* = 3 slices, *N* = 1 mouse. Yellow arrowheads indicate giant (>1 μm diameter) corticothalamic boutons. Scale bar, 5 μm. **l** *Top*: Two optical pulses (@5 Hz, 3 ms, blue) induced paired-pulse depression of the EPSPs (average of 40 trials). *Bottom*: Voltage-clamp recording of EPSCs (@20 Hz, 3 ms) reveal postsynaptic depression (trace average of 4 trials). POm neurons held at −76 mV (Liquid junction potential corrected). **m** Population data of peak EPSC amplitude reduction (10th /1st EPSC) revealed a frequency-dependent depression. One-way ANOVA *P* = 0.0079. Tukey's multiple comparisons test 20 vs. 10 Hz, *\*P* = 0.03, 20 vs. 1 Hz, *\*\*P* = 0.009, 10 vs. 1 Hz *P* = 0.83. *n* = 7 POm neurons from *N* = 6 mice. Data are presented as mean ± SEM. **n** Mouse brain schematic (adapted from[102]) with experimental approach and example in vivo juxtacellular recordings (*V*Juxta) of L5 (*top*) and POm neurons (*bottom*), showing LED-evoked (blue bars, @1 Hz, 5 pulses per trial) and spontaneous spiking. **o** High magnification of a L5 AP burst (200 Hz) temporally aligned with delayed single AP in the POm. Source data are provided as a Source data file. Atlas diagram in (**b**) adapted from ref. 103.

Interestingly, Ctrl spiking showed two distinct peaks in the histogram (Fig. 2i). The first peak may correspond to monosynaptic L5→POm timing delays[27], rendering the possibility that the second peak reflects spiking through polysynaptic connections (L5→L5→POm). To explore this possibility, we split the Ctrl cells into putative mono- and polysynaptic connections based on the distribution of their delay times (distribution in Supplementary Fig. 2b). We used the mean of each group as an objective cutoff to split the populations (Ctrl, 11.30 ms and Cpz, 17.35 ms). The putative monosynaptic population in the control group matched the published delay for L5 to POm connections[27]. However, we did not find a significant interaction effect of connection type on delay or variance of spiking (Supplementary Fig. 2e, f). Both mono- and polysynaptic connections were delayed by 5 ms, suggesting the slowed conduction is determined by the myelin loss of the L5 to POm axonal path. Taken together, these results indicate that demyelinated long-range corticothalamic axons show large temporal variance and are delayed by multiple milliseconds.

**Reduced spike transmission during high-frequency bursts**

A canonical finding for demyelinated axons is the increased frequency-dependent failures of AP propagation and conduction block[20,36]. Previously, Cpz treatment was found to increase in vitro L5 PN burst firing evoked with somatic-current injections[19]. The optotagged juxtacellular recordings showed, however, that with optogenetically evoked responses there was no significant difference in spiking probability in either L5 or POm region (Fig. 3a–d). Further analysis showed that evoked responses in L5 were unchanged: at high light powers (>5 mW) L5 neurons fired ~3 spikes at instantaneous frequencies around 200 Hz (Fig. 3b, c, Supplementary Fig. 3a). In contrast, juxtacellular recording in the POm revealed trends for a reduction in probability (Fig. 3d) and the average number of spikes (-1.3 spikes, *P* = 0.053), but without change in the instantaneous frequency (-150 Hz, Supplementary Fig. 3b).

One caveat in the analysis of spike transmission probability is the strong frequency-dependent synaptic depression of the giant L5–POm synapse, typically translating L5 bursts of 3–4 APs from multiple neurons into 1–2 APs in the POm[22,24,27]. In other words, while the first L5 spike in the burst gets reliably transmitted at high probability, most of the subsequent depolarizations at the giant terminal will not translate into POm spikes, despite a reliable spike propagation from the AIS to the giant terminal. Due to this filtering, detection of transmission failure along the demyelinated axon based on somatic recordings alone is challenging. Even when some presynaptic spikes fail, the remaining ones could still drive the POm neuron to spike threshold (Fig. 3b, d). To obtain a more detailed insight into putative failure

of spike transfer within the burst code, we made additional, temporal analyses to detect which spikes within the presynaptic cortical burst are reliably transmitted. We overlaid all spikes recorded in POm neurons for a given light intensity (25 mW) and temporally aligned them to reveal the optogenetically evoked first and second spike in the presynaptic burst (Fig. 3e). The analysis of spike probabilities within these windows revealed a significant reduction of POm spike probability for the spike in the first cluster (from ~90 to ~70%, Fig. 3f), indicating an increased failure for the first spike in a burst. In addition, also the delay from light onset to the first spike was increased (Supplementary Fig. 3c), indicating that the demyelination-induced temporal delay most likely reflects a decrease in CV and is independent of AP failures. In addition, the variance of the first spike was significantly increased (Supplementary Fig. 3d), indicating that the observed overall increase in variance (Fig. 2g) was caused by both first-spike failure, leading to the onset times jittering between the first and the second spike, as well as jittering within first-spike arrival times (c.f. Supplementary Figs. 2c and 3d). Thus, although the overall low corticothalamic spike transfer of ~3 cortical spikes into 1 POm spike is unchanged on average, failures occur selectively for the first POm spike. In addition, we analyzed a subset of recordings in the thalamus (2 out of 3 per group located in POm), in which whole-cell access was obtained (Fig. 3g). In these recordings, we could detect subthreshold EPSPs and averaged the responses from light-evoked trials without APs (Fig. 3h). To compare the time course of the mean EPSP response, we averaged the voltage transients in 5 ms bins from the onset of the EPSP. This analysis revealed significantly lower EPSP amplitudes in Cpz mice (Fig. 3i). In addition, when we normalized the data to the first EPSP, we found that while control EPSPs showed rapid summation followed by depression, in Cpz responses were only summating slowly (Fig. 3j).

The demyelination-induced loss of rapid L5–POm transmission could be caused by multiple non-mutually exclusive mechanisms. The first explanation may be that the synaptic properties of L5–POm boutons or the membrane properties of the POm neurons are impaired due to demyelination locally in the POm. To study the boutons directly we stimulated ChR⁺ fibers in the POm at different frequencies and measured the excitatory postsynaptic currents (EPSCs) in voltage-clamp mode in acute in vitro brain slices (Supplementary Fig. 4a). The results showed that the maximum amplitude and frequency-dependent adaptation of the EPSCs were similar after Cpz treatment (Supplementary Fig. 4b–e). Furthermore, immunofluorescence analysis showed the size of the giant boutons was not changed (Supplementary Fig. 4f). Finally, we investigated the intrinsic membrane excitability of POm neurons by in vitro patch-clamp recordings. Comparisons revealed no differences in the resting membrane

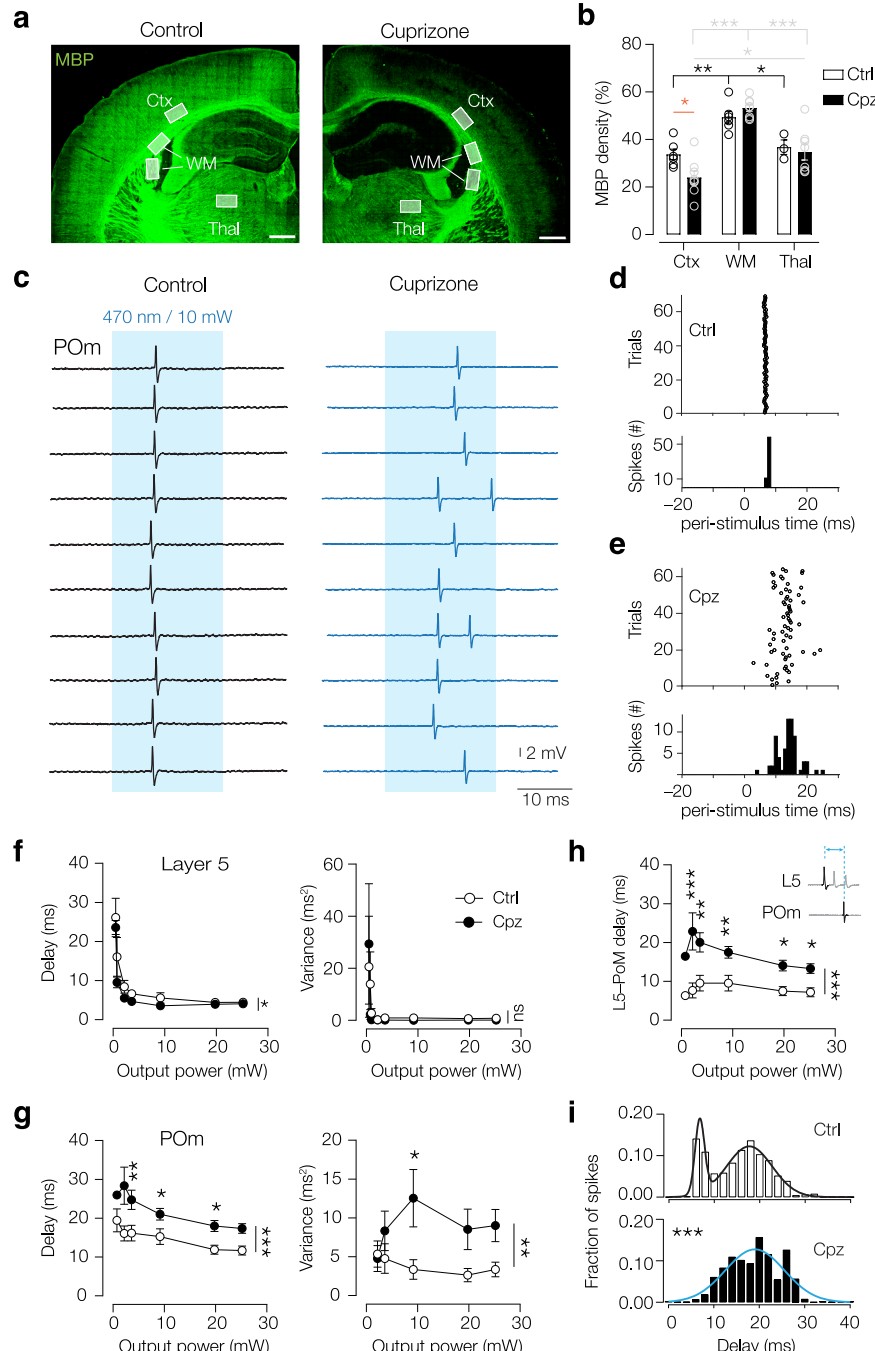

**Fig. 2 | Cortical demyelination is associated with impaired L5–POm spike propagation. a** Confocal image of MBP (green) in coronal slices of control (Ctrl) and cuprizone (Cpz), revealing regional demyelination. White shaded boxes indicate representative ROIs for analysis. Scale bars, 500 μm. **b** MBP coverage significantly reduced in the cortex (Ctx) but not in the white matter (WM) or thalamus (Thal). Two-way ANOVA P = 0.2 treatment; *P* < 0.001 region; *P* = 0.04 interaction. Tukey's multiple comparisons test Ctrl vs Cpz (orange), within Ctrl (black), within Cpz (gray). *P < 0.05, **P < 0.01, ***P < 0.001. *n* = 3 ROIs/animal, *N* = 6 mice (Ctrl Ctx, WM), 4 mice (Ctrl Thal), 8 mice (Cpz Ctx, WM, Thal). For exact *p* values see supplementary information. **c** Optically evoked (blue shading) spikes in 10 consecutive trials in Ctrl (*left*) and Cpz (*right*) POm neuron. Raster plot and peri-stimulus time histogram of opto-evoked spiking show precise and fast spike timing in Ctrl neurons (**d**), whereas spiking is delayed and jittered in Cpz neurons (**e**). **f** Onset delays and variance of spiking were high at low output power but decreased at higher light intensities. Cpz treated neurons (closed circles) spiked on average -1.5 ms earlier compared to control (open circles); 2-way ANOVA (delay) *P* < 0.0001 output power; *P = 0.0198 treatment; *P* = 0.7032 interaction. (variance) *P* = 0.0174 output power;

*P* = 0.764 treatment; *P* = 0.9373 interaction. *n* = 7 neurons, *N* = 4 mice (Ctrl), *n* = 5 neurons, *N* = 4 mice (Cpz). **g** Delay (*left*) and variance (*right*) of POm spiking was significantly increased after demyelination, independent of light output power. Two-way ANOVA delay *P* < 0.0001 output power; ***P* < 0.0001 treatment; *P* = 0.6142 interaction. Variance *P* = 0.7939 output power; **P* = 0.0059 treatment; *P* = 0.55536 interaction. **h** Delay time from L5 to POm spiking was significantly increased after demyelination. Two-way ANOVA ***P < 0.0001 treatment; *P* = 0.0305, output power; *P* = 0.264 interaction. **g**, **h** *n* = 19 neurons, *N* = 13 mice (Ctrl, open circles), *n* = 18 neurons, *N* = 11 mice (Cpz, closed circles). **i** Histogram of all onset spikes of all cells (25 mW stimulation) showed two peaks for Ctrl neurons (white, top), best fit by a sum of 2 Gaussian fits (black line, early and late responses, 6.8 and 17.7 ms, *r*² = 0.98, sum of squares 0.00098), but a single Gaussian fit for Cpz (blue line, 19.03 ms, *r*² = 0.84, sum of squares 0.0086). Kolmogorov–Smirnov test ***P < 0.0001, *n* = 15 neurons, *N* = 13 mice (Ctrl), *n* = 17 neurons, *N* = 11 mice (Cpz). For all multiple comparisons, Šídák's multiple comparisons test, *P* > 0.05, *P < 0.05, **P < 0.01, ***P < 0.0001. All data are plotted as mean ± SEM. Source data are provided as a Source data file.

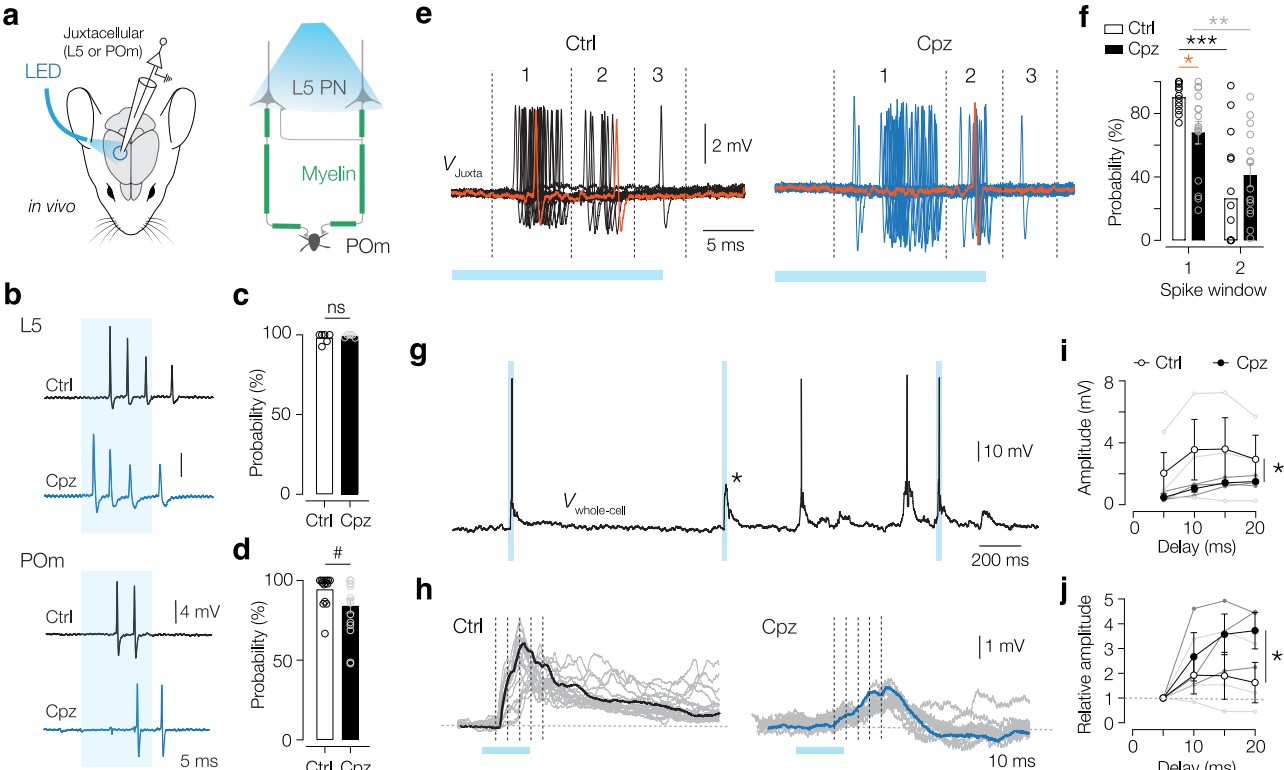

**Fig. 3 | Myelination is critical for L5–POm burst transfer. a** Schematic of in vivo juxtacellular recordings in L5 or POm (mouse brain, adapted from ref. 102), combined with optogenetic activation of L5–POm axons. **b** Optically evoked (blue shading) burst firing in L5 or POm neurons in Ctrl (black) and Cpz mice (blue). **c** Probability of optically evoked firing in L5 was similar (25 mW). Mann–Whitney test $P = 0.92$. $n = 6$ neurons, $N = 4$ mice (Ctrl) and $n = 5$ neurons, $N = 4$ mice (Cpz). **d** Probability of optically evoked firing in POm (25 mW) showed a trend of being reduced. Mann–Whitney test #$P = 0.0785$. $n = 15$ neurons, $N = 13$ mice (Ctrl), $n = 16$ neurons, $N = 11$ mice (Cpz). **e** Overlay of 39 (Ctrl) or 50 (Cpz) opto trials depicting the clustering of spikes into temporally separate windows (numbered, first, second, and third spike in a burst). Orange trace highlights one individual trial. **f** Reduced probability of a POm spike in the first temporal window (corresponding to the first spike in a burst). Repeated measures 2-way ANOVA, $P = 0.3527$ treatment;

$P < 0.0001$ spike nr; $P = 0.0564$ neuron; $P = 0.03$ interaction. Uncorrected Fischer's LSD *$P = 0.0395$, **$P = 0.0028$, ***$P < 0.0001$. $n = 12$ neurons, $N = 9$ mice (Ctrl), $n = 15$ neurons, $N = 9$ mice (Cpz). **g** Example whole-cell recording with evoked (blue) and spontaneous supra- and subthreshold EPSPs (asterisk). **h** Individual (gray) and averaged EPSP responses (bold) for Ctrl and Cpz neurons in response to optical stimulation (blue bar). Dotted lines indicate 200 Hz event window. **i** EPSP amplitudes significantly reduced in Cpz. Two-way ANOVA *$P = 0.043$ treatment; $P = 0.75$; $P = 0.97$ interaction. Šídák's multiple comparisons test $P > 0.05$ for all comparisons. $n = 3$ neurons, $N = 3$ mice (Ctrl and Cpz). **j** Relative EPSP amplitudes, normalized to the first 5 ms window, show increased summation in Cpz. Two-way ANOVA *$P = 0.045$ treatment; $P = 0.105$ time; $P = 0.5$ interaction. Šídák's multiple comparisons test $P > 0.05$ for all comparisons, $n = 3$ neurons, $N = 3$ mice (Ctrl and Cpz). All data are plotted as mean ± SEM. Source data are provided as a Source data file.

potential (RMP), input resistance, nor the current/voltage thresholds for AP generation (Supplementary Fig. 4g–j). In addition, the current–frequency responses were not changed (Supplementary Fig. 4k, l). The fraction of POm neurons that responded with bursting to hyperpolarizing or depolarizing current steps was not different between Cpz and Ctrl (Supplementary Fig. 4m). Together, these results indicate that neither the presynaptic L5 giant synapse properties nor the postsynaptic POm membrane properties can explain the impaired L5–POm spike transfer probability induced by demyelination.

**Neuropixels recordings reveal increased delays and spike failures between putatively connected L5–POm pairs**
A second possibility is that following demyelination an altered (default) network activity affects the neuronal excitability of POm neurons. Under anesthesia, the corticothalamic spike transfer is powerfully modulated by cortical up- and down states[27]. To determine the spontaneous ongoing activity in S1 and POm, we performed high-density recordings with Neuropixels probes, inserted at a 30° angle to target both regions with one probe (Fig. 4a–c). The results showed no change in spontaneous firing frequency in POm neurons (-0.6 Hz) or L5 neurons (-0.5 Hz, Fig. 4d, e), nor in the intermediate brain regions recorded along the shaft (CA1, CA3, S1, Supplementary Fig. 5a–c). Furthermore, these findings were independent of the depth of

anesthesia (Supplementary Fig. 5d). Next, we analyzed whether there were differences in burst firing (spike events of 2 or more APs at >100 Hz). The results showed the fraction of bursting units (minimally 10 bursts during spontaneous activity recordings) was constant (Fig. 4f, g). Interestingly, however, in Cpz mice we observed a significantly increased fraction of burst events relative to all spikes in L5 neurons as well as burst frequency (Fig. 4h, j). Consistent with the findings with juxtacellular recording and in vitro whole-cell recordings, in the POm neither the fraction of bursting units, the instantaneous frequency of spontaneous bursts, nor the fraction of burst events, was affected (Fig. 4g, i, k). These findings further exclude the possibility that a change in POm excitability accounts for the observed impairment in evoked burst transmission. Next, since the Neuropixels probe recorded from L5 and POm neurons simultaneously, we analyzed the data for putative connected pairs. In both Ctrl and Cpz mice we found a subset of POm neurons that significantly increased their firing probability in tight temporal alignment with L5 single spikes or bursts (Fig. 4l, m), indicating connected pairs. While the connection probability was unchanged by Cpz treatment, the delay from L5 to POm spiking was significantly increased (Fig. 4n, o). Importantly, the delay between L5 and POm spiking closely matched the values we measured with juxtacellular recordings identified as putative monosynaptic connections (relative to L5 spike ~3 ms in Ctrl vs ~8 ms in Cpz).

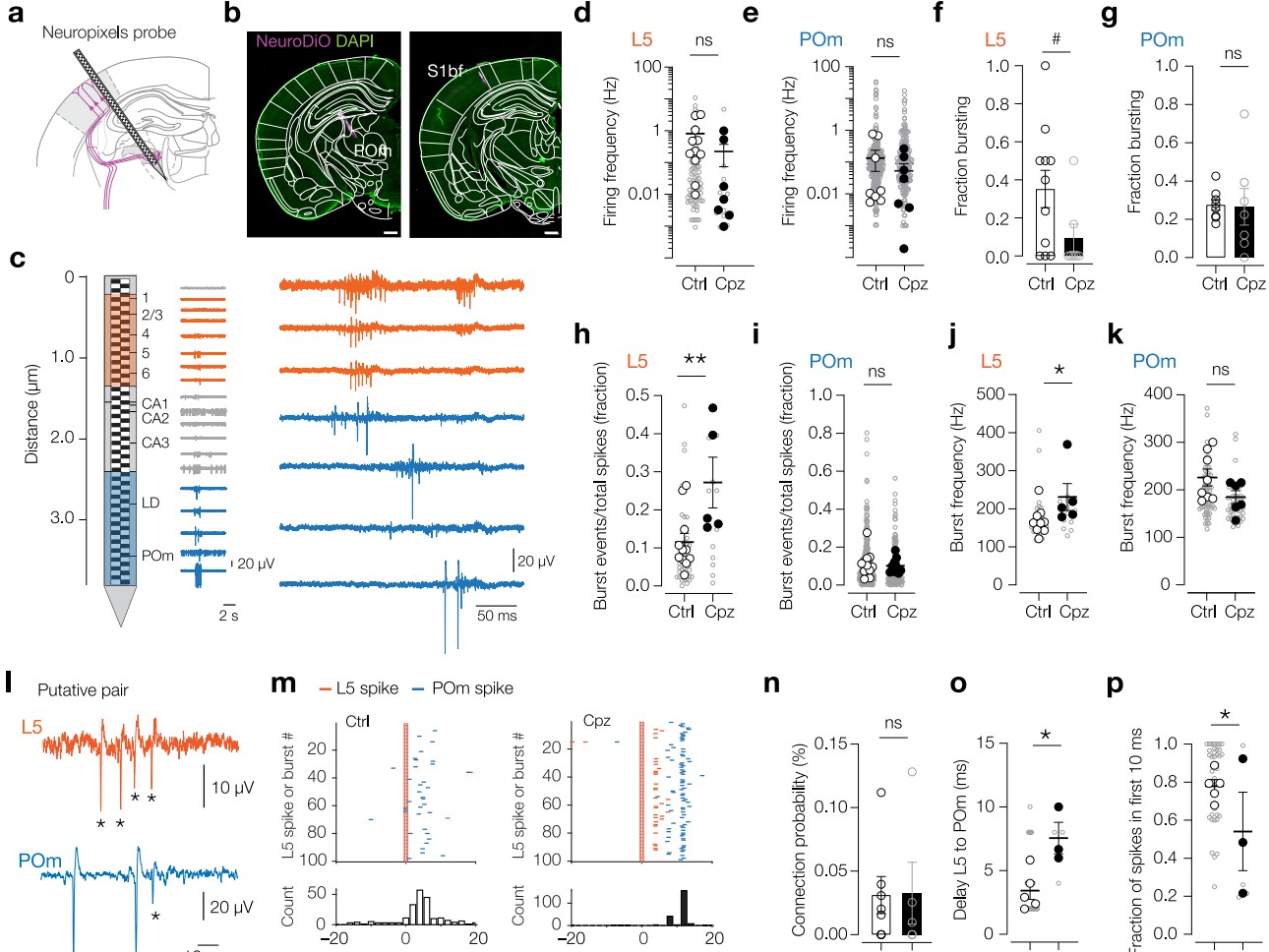

**Fig. 4 | Cuprizone-induced L5 burst firing associated to delayed and low POm spike rates. a** Schematic of in vivo Neuropixels recordings from S1bf to POm. **b** NeuroDiO (magenta) coated probes for reconstruction of the 3D trajectory using DAPI (green) and atlas diagrams (−2.30 and −1.94 mm from bregma, adapted from ref. 103). Scale bars, 500 μm. **c** *Left:* the probe captured activity from single units in the cortex (orange), hippocampus and thalamus (blue). *Right:* Example voltage-time single and multi-units captured in both S1 and POm. **d** Average firing frequency of L5 single units not affected by demyelination. Mann–Whitney test over animal means *P* = 0.065. *n* = 73 neurons (gray open circles), *N* = 11 mice (white filled circles, Ctrl), *n* = 14 neurons, *N* = 7 mice (black filled circles, Cpz). **e** Average firing frequency of POm single units. Mann–Whitney test over animal means *P* = 0.54. *n* = 277 neurons (gray open circles), *N* = 8 mice (white filled circles, Ctrl), *n* = 190 neurons, *N* = 7 mice (black filled circles, Cpz). **f** Fraction of bursting units in L5 showed a trend to reduce. Unpaired *t*-test *P* = 0.078, *t* = 1.887, *N* = 11 mice (Ctrl), *N* = 7 mice (Cpz). **g** Fraction of bursting units in POm was -25% in both groups. Unpaired *t*-test *P* = 0.91, *t* = 0.118, *N* = 8 mice (Ctrl), *N* = 7 mice (Cpz). **h** Fraction of burst events over all spikes in L5 increased after demyelination. Mann–Whitney test over animal means, **P* = 0.013, *n* = 41 neurons, *N* = 11 mice (white filled circles, Ctrl), *n* = 14 neurons, *n* = 5 mice, black filled circles, Cpz. **i** Fraction of burst events over all spikes in POm was unchanged. Unpaired *t*-test over animal means *P* = 0.87, *t* = 0.167, *n* = 375 neurons, *N* = 12 mice (white filled circles, Ctrl), *n* = 168 neurons,

*n* = 9 mice (black filled circles, Cpz). **j** Instantaneous burst frequency in L5 increased after demyelination. Mann–Whitney test over animal means **P* = 0.019, *n* = 41 neurons (gray open circles), *N* = 11 mice (white filled circles, Ctrl), *n* = 14 neurons, *N* = 5 mice (black filled circles, Cpz). **k** Instantaneous burst frequency in POm was unchanged. Unpaired *t*-test over animal means *P* = 0.11, *t* = 1.744, *n* = 72 neurons (gray open circles), *N* = 8 mice (white filled circles, Ctrl), *n* = 44 neurons, *N* = 6 mice (black filled circles, Cpz). **l** Spontaneous, temporally aligned burst firing (spikes within burst indicated with asterisks) of putatively connected pairs of L5 (orange) and POm (blue) single units. **m** Example raster plots (top) and spike cross-correlograms (bottom) of POm neurons (blue raster, Ctrl left and Cpz right) showing an increased spiking, temporally aligned to L5 spikes (orange line), or L5 bursts (orange raster), indicating an identified coupled pair. **n** Connection probability (percentage of putatively connected pairs from the number of possible pairs) was unchanged. Mann–Whitney test *P* = 0.72, *N* = 7 mice (Ctrl), *N* = 5 mice (Cpz). **o** Delay between spikes in putatively connected L5⇢POm pairs (spike to 1st spike). Nested *t*-test ***P* = 0.038, *t* = 2.657. *n* = 38 pairs, *N* = 5 mice (Ctrl), *n* = 5 pairs, *N* = 3 mice (Cpz). **p** Fraction of spikes in POm neuron in the first 10 ms in the 20 ms following a spike in the putatively connected L5 neuron. Nested *t*-test **P* = 0.015, *t* = 2.544. *n* = 38 pairs, *N* = 5 mice (Ctrl), *n* = 5 pairs, *N* = 3 mice (Cpz). All data are plotted as mean ± SEM. Source data are provided as a Source data file.

Furthermore, to examine the increased failure of the first spike in a burst (Fig. 3f) with the identified pairs, we analyzed the fraction of spikes within the first and second 10-ms windows relative to the first POm spikes (Fig. 4p). The data showed a significantly reduced fraction of spikes in the first cluster (c.f. Fig. 3f), providing additional and independent support of an impaired L5–POm spike transfer. In summary, the Neuropixels recordings show that demyelination leads to an increased bursting in L5, but not POm, excluding the possibility that

network excitability changes account for the reduced L5-evoked spiking in the POm.

### Diverse morphological changes contribute to delayed corticothalamic feedback

A third possibility explaining the impaired L5–POm spike transmission may be a failure in AP propagation along the demyelinated axon. Cuprizone induces frequency-dependent AP failures[19,20]. Myelin loss

with cuprizone treatment is highly region- and cell-type dependent[34,35,37] but how myelination of corticothalamic axons is affected remains unknown. To determine the precise myelin pattern, we 3D-reconstructed individual myelin sheaths as well as the flanking paranodal domains (Caspr[+]) along genetically labeled L5–POm axons (Fig. 5a, b, Supplementary Fig. 6a). L5–POm axons exhibited robust demyelination in the cortex with axons either completely or partially lacking myelin. Accordingly, there was a significantly lower percentage of axonal stretches covered by myelin, and myelinated internodes were significantly shorter in Cpz mice (Fig. 5c, d). Within the thalamus we also observed a lower myelin coverage along the L5 axons (Fig. 5b, c). Due to the high density of myelin fluorescence in the white matter, we could not resolve individual internodes in this region. In line with demyelination, however, there was also a loss of Caspr expression enabling indirect assesment of myelin loss. Quantitative comparison of the three regions showed the density of Caspr[+] puncta was significantly reduced in the infragranular layers of the cortex (Ctx) but not in the white matter (WM) nor in the thalamus (Thal) (Fig. 5e, f), in accordance with previously published results[38]. Whereas in Ctrl L5 axons, myelin sheaths were always flanked by two paranodes, in Cpz mice sheaths lacked either one or both flanking paranodes (Fig. 5g). These paranodes appeared to be redistributed along the axon with gaps appearing between myelin onsets (white arrows) and Caspr[+] puncta (yellow arrows, Fig. 5h).

Another noticeable morphological alteration of L5 Cpz axons was the appearance of axonal swellings or "spheroids" in the Ctx and WM (Fig. 5i, j), which have been described to occur after demyelination in animal models and MS patients[39–41]. We further studied the disarrangement of nodal domains using 3D reconstructions (Supplementary Fig. 6d). Consistent with the demyelination pattern (Fig. 2b), the Ctx contained the largest fraction of aberrant nodes, including partial loss of nodal compartments, overlapping domains or gaps (Supplementary Fig. 6d, e). On average, the AnkG[+] regions of the nodes were significantly increased after demyelination, exclusively in the Ctx (Supplementary Fig. 6f). To investigate the nodal domains in more detail, we 3D-reconstructed individual axons from the three subregions and measured the AnkG[+] distances (Supplementary Fig. 6g, h). The data showed a significant increase of node-to-node distance in the Ctx, but not in the other regions (Supplementary Fig. 6e). In addition, we reconstructed the AIS, to follow the order of nodes along proximal axons (Supplementary Fig. 6i). When analyzing the distribution of nodes as a function of node number (counting from the soma), we found that already the first node was significantly further away from the start of the axon (Supplementary Fig. 6i), while AIS length was not changed (Supplementary Fig. 6j). Together, these findings indicate that the proximal region of the L5–POm projection is strongly affected by cuprizone-induced demyelination.

## Model simulations reveal gray matter myelin loss impedes burst propagation

To examine the relative contributions of region-dependent myelin loss, node loss and spheroids to changes in AP propagation we made a computational model of a L5 pyramidal neuron with a corticofugal axon projecting into the POm region ("Methods," Fig. 6a). The morphology was based on mCherry[+] neurons and fluorescent markers guiding the anatomical reconstruction of paranodal, nodal and internodal domains (Supplementary Fig. 7a, b). A stitched 3D morphology representing a single L5–POm projection was imported into NEURON and converted into a conductance-based multicompartmental model in which the myelin sheath was computationally integrated as a double-cable circuit[3] (Supplementary Fig. 7c). The active model reproduced APs initiating in the AIS and backpropagating into the dendrite (Fig. 6b, c). To simulate experimentally realistic forward propagation, we plotted the CV (path length AIS to POm terminal/time) against myelin sheath thickness between 0 and 21 lamellae (by varying radial myelin

resistance ($R_{my}$) and capacitance ($C_{my}$), see "Methods"). The CV exponentially increased with myelin thickness, such that 9 lamellae predicted a CV of 1.93 m s$^{-1}$ (Fig. 6b, Supplementary Movie 2). Notably, a myelin thickness of 9 lamellae is in range of the myelin ultrastructure in mouse corpus callosum and thalamus (7–9 lamellae[31,42,43]) and the in vivo recorded CV for cortico-POm axonal tracts is 1.82 m s$^{-1}$ (ref. 44). Assuming a nominal delay of 1.5 ms between the pre- and postsynaptic AP, accounting for glutamate transmitter release and charging of the postsynaptic membrane to AP peak rate (this study, and ref. 27), the spike-to-spike delay with 9 lamellae was 3.74 ms, well in range of in vivo juxtacellular and Neuropixels recordings (on average 3.13 and 3.79 ms, respectively, Fig. 4, Supplementary Fig. 7d, e).

Next, we explored the biophysical consequences of myelin pathologies in cortex, white matter and thalamic regions. Removing the conductive myelin sheath in the model predicted slow CV values quantitatively in agreement with experimental data (Fig. 6d, Supplementary Fig. 8a, b). Local diameter changes of the axon, simulating axonal swellings, caused minor velocity changes but became substantial when the myelin sheath was computationally decompacted (Supplementary Fig. 8c, d). Next, when we removed the myelin sheath in the Ctx we observed that the spike arrived in the POm with an additional delay of ~1 ms and further slowed by removing the nodes of Ranvier (2.7 ms delay). When we abolished the myelin layer and nodes also in the POm the spike delayed by ~4.7 ms. In contrast, removing myelin and nodes from the WM caused delays inconsistent with experimental data (13 and 17 ms, respectively, Supplementary Fig. 8e, f). These results indicate the model recapitulated a plausible observed range of spike delays and support the observation that Cpz treatment mostly affects the cortex and POm, consistent with the observed myelin loss and sparse nodal damage in these regions (Fig. 5).

Finally, to test the sole contribution of myelin to AP burst transfer from L5 to POm under in vivo-like activity, we added to the deterministic model randomly fluctuating synaptic potentials resembling in vivo inputs[45]. By additionally injecting a 20-ms duration current into the soma, we triggered 3–4 APs similar to the optogenetically evoked burst (1.7 nA current injection, Fig. 6e). The control model neuron produced AP bursts at an average frequency of ~190 Hz ($n = 72$ spikes from 20 trials). Running simulations without myelin in cortex and POm (but intact nodes) did not change the average number of evoked APs (Ctrl, on average 3.6 vs. Cpz, 3.8 spikes, two-tailed $P = 0.31$, $n = 20$ runs) but the intra-burst frequencies were significantly higher (Ctrl, 190.4 ± 7.57 Hz; Demyelinated, 219.4 ± 10.53 Hz, paired $t$-test $P = 0.0317$), in line with the Neuropixels data (Fig. 4j). Removing myelin also produced large temporal dispersion of L5–POm delays (variance 2.87 vs 18.47 ms$^2$) and a nearly 2-fold slowing of average CV between L5 and POm (Fig. 6f). Importantly, demyelination increased the failure of individual APs within the burst from ~6 to ~32% (two-sided Fisher's exact test, $P < 0.0001$ Fig. 6) which upon closer inspection of the model occurred beyond the AIS, around the first node of Ranvier (Supplementary Movie 2). To examine whether the increased failure was caused by the higher frequencies at which bursts were initiated at the AIS, we plotted AIS frequency versus POm pre-synapse frequency. These data showed that in the absence of the myelin sheath burst transfer was hampered from frequencies as low as ~130 Hz, substantially lower compared to frequency of failures in the control axon, excluding that L5–POm failures are caused by the increased AIS output frequency (Fig. 6g, Supplementary Fig. 8e). Together, the computational simulations demonstrate that gray matter myelin loss at the L5–POm axonal projection causes failure in the propagation of high-frequency spikes within a burst.

## Precise temporal integration of multiple inputs requires L5 myelination

If cortical drive of the POm by L5 bursts is weakened this may impact on the coincidence detector function of the POm. To test this

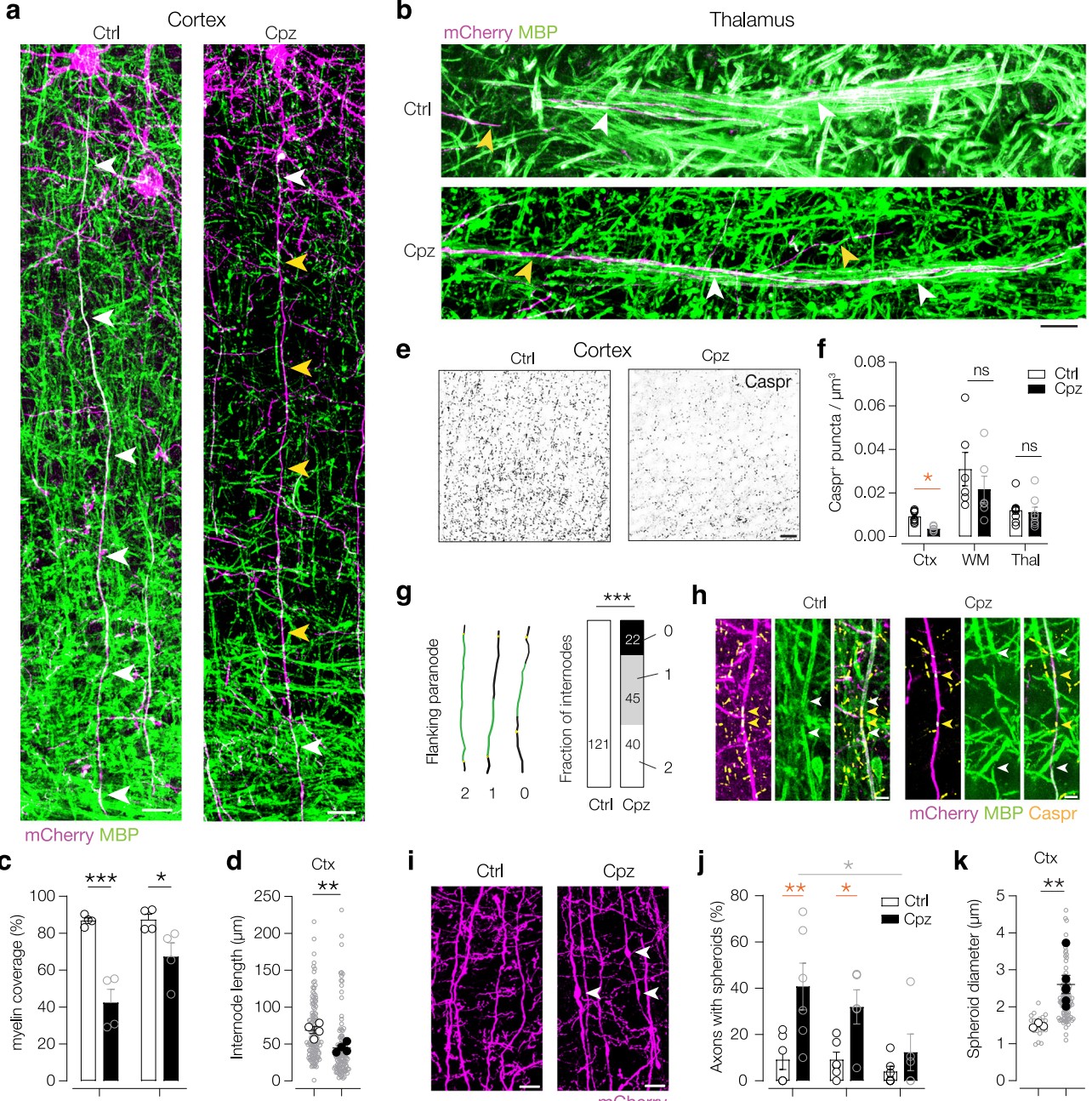

**Fig. 5 | Cuprizone-induced demyelination is limited to gray matter regions of L5–POm axons.** Confocal immunofluorescence images of mCherry (magenta) and myelin (MBP, green) in cortex (**a**) and thalamus (**b**) were used for 3D colocalization analysis. White arrows point to mCherry⁺ MBP⁺ regions, yellow arrows point to mCherry⁺ demyelinated axons. Scale bars 20 μm (**a**), 15 μm (**b**). Data replicated in *N* = 4 mice (Ctrl and Cpz). **c** Population data of myelin coverage in Ctx and Thal revealing a significant myelin loss after Cpz treatment. Two-way ANOVA *P* < 0.0001 treatment; *P* = 0.0376 region; *P* = 0.0451 interaction. Šídák's multiple comparisons Cpz vs. Ctrl for Ctx ***P* = 0.0002 and Thal, **P* = 0.0456, *n* = 58 internodes (Ctx Ctrl), *n* = 75 (Ctx Cpz), *n* = 70 (Thal Ctrl), and *n* = 112 (Thal Cpz) from *N* = 4 mice (Ctrl and Cpz). **d** Internode length decreased. Nested *t*-test ***P* = 0.0096, *t* = 3.74, *n* = 133 internodes (Ctrl, gray open circles), *n* = 110 (Cpz), *N* = 4 mice (filled circles). **e** Example confocal image of paranodes (Caspr⁺) in L5/6 of the cortex. Note the loss in Cpz. Scale bars, 50 μm. **f** Caspr density significantly reduced in Ctx but not WM or Thal. Kruskal–Wallis *P* < 0.0001, orange between treatments, multiple comparisons two-stage linear step-up procedure of Benjamini, Krieger, and Yekutieli **P* = 0.0254.

*N* = 9 mice (Ctrl Ctx and Thal), *N* = 6 (Ctrl and Cpz WM), N = 7 (Cpz Ctx), N = 8 (Cpz Thal). **g** In control axons, internodes were always flanked by two paranodes (white). In Cpz axons, internodes frequently lacked one (gray) or both (black) paranodes. Fischer's exact test ****P* < 0.0001. *n* = 121 internodes from *N* = 4 mice (Ctrl). *n* = 107 internodes from *N* = 4 mice (Cpz). **h** Example images of bare stretches of axons between nodal regions and internodes. White arrows, myelin ending and yellow arrows, Caspr⁺ paranodes. Scale bars, 5 μm. *N* = 4 mice (Ctrl and Cpz). **i** Immunofluorescence images of axonal spheroids after demyelination. Scale bars, 10 μm. **j** Percentage axons with spheroids in Ctx, WM and Thal significantly increased. Two-way ANOVA *P* = 0.0004 treatment; *P* = 0.0369 region; *P* = 0.1993 interaction. Šídák's multiple comparisons, orange between treatment, **P* = 0.0192 (WM Ctrl vs Cpz), **P* = 0.0118 (Cpz Ctx vs Thal), ***P* = 0.0012, *N* = 6 mice (Ctrl), *N* = 5 mice (Cpz). **k** Size of swellings in the Ctx was significantly increased. Nested *t*-test ***P* = 0.0081, *t* = 3.65, *n* = 22 swellings (gray open circles), *N* = 3 mice (white filled circles, Ctrl), *n* = 106 swellings, *N* = 6 mice (black filled circles, Cpz). All data are plotted as mean ± SEM. Source data are provided as a Source data file.

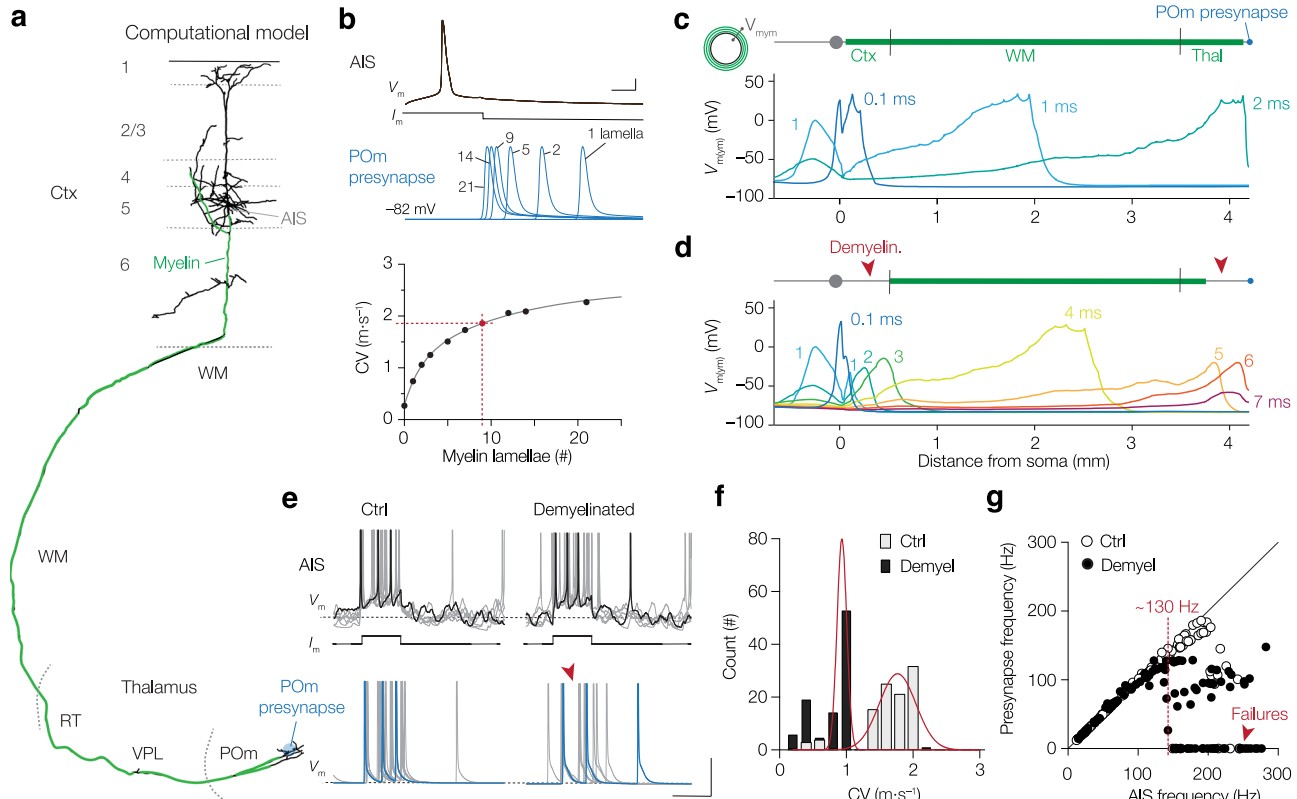

**Fig. 6 | Computational modeling indicates cortical myelin loss impedes high-frequency burst propagation. a** Morphology of the compartmental model. Axon spans ~4.2 mm from the L5 soma to the giant bouton in the POm, including biologically realistic compartments of myelinated internodes (green) and nodal domains. For details see Supplementary Fig. 7. **b** Somatic-current injection evoked single APs initiated in the AIS. Increasing myelin lamella number speeds CV (fit double exponential to 10 model settings). Red dotted line indicates 1.9 m s$^{-1}$ at 9 myelin lamellae. **c** Space plot of neuronal ($V_m$) voltage profiles across soma and dendrites and transmyelin potentials ($V_{mym}$) along the axons coursing along cortex (Ctx), white matter (WM) and thalamus (Thal). Note the sharp transients at the nodes of Ranvier. **d** Space plot profiles of $V_{mym}$ and $V_m$ of a partially demyelinated axon (red arrows). Eight distinct time points are differentially colored and overlaid.

Note the long duration to propagate within the cortex (3 ms). See also Supplementary Movie 2. **e** *Left*, seven overlaid AP bursts during in vivo-like random synaptic inputs. Voltage traces from AIS (black) and giant presynaptic terminal in POm (blue) evoked with a 20-ms rectangular current injection. Note the spike failure in the high-frequency cluster in the demyelinated model (red arrow). **f** Histogram of L5–POm CVs. Data from 40 trials and spike intervals binned with 0.2 m s$^{-1}$. Non-linear Gaussian fits (red); peak 1.77 and 0.93 m s$^{-1}$ (S.D. 0.27 and 0.07, $n = 104$ Ctrl and $n = 83$ Demyel., respectively). **g** Demyelination caused AP propagation to fail from a frequency of ~130 Hz (demyelinated, $n = 122$ spike intervals, 40 trials, closed black circles) in comparison to a control frequency of ~200 Hz ($n = 109$ spike intervals, 40 trials, open circles). Black line shows the identity line. Source data are provided as a Source data file.

conjecture, we employed dual optogenetic and whisker stimulation at varying delays (Fig. 7a, b). First, we tested whisker stimulation alone to determine the spiking response upon whisker stimulation (Fig. 7c, Supplementary Fig. 9a). Interestingly, we observed a wide range of responses: Some neurons responded with strong, immediate spiking within a few milliseconds after stimulus onset ("early," Fig. 7c). Other neurons showed increased spiking between 100 and 300 ms after stimulus onset ("late"), were inhibited by whisker stimulation ("inhibited") or showed no response at all (Fig. 7d). These observations are consistent with previous studies, characterizing whisker responses in POm under anesthesia[26,46]. When comparing the distribution of responsive neurons, there was no significant difference between Cpz and Ctrl (Fischer's exact test $P = 0.56$, Fig. 7d). Interestingly, there was no difference in the delay between whisking and POm spiking recorded in the "early" neurons (including neighboring nuclei), indicating the direct and myelinated sensory input pathway via the trigeminal nucleus spinals interpolaris (Sp5i) remained unaffected by cuprizone treatment (Supplementary Fig. 9a).

We next paired optogenetic and whisker stimulation (Fig. 7a, e, f). Spiking probability in POm is increased supralinear when L5 and whisker feedforward input are combined[24]. Importantly, for each cell there was a different "integration window" during which

probabilities were significantly increased. Peak response times ranged from negative delays (opto before whisker) to positive delays (whisker before opto), with a bias towards positive delays. We therefore tested different relative delays ranging from −50 ms to 50 ms in steps of 10 ms and reduced the laser power and piezo strength to low light and amplitudes, respectively, such that the response probability of stimulation alone was low (opto to whisker delay, o−w, Fig. 7e, f). Interestingly, we found cells that had significantly increased as well as cells with a decreased probability of spiking (relative to the sum of spikes in opto and whisker alone trials, Fig. 7f, Supplementary Fig. 9b). The distribution of cell response types was not different between Ctrl and Cpz (Fig. 7g), neither was the relative fold-change of the significant responses (Fig. 7h). The delay at which the strongest change occurred (peak delay) was not different between treatment groups for the cells with increased probability, indicating that demyelination did not affect the temporal distribution of coincidence encoding within the population (Fig. 7i). However, the width of the integration window, as measured by the sum of all tested opto to whisker delays at which significant firing changes occurred, was significantly increased for Cpz cells (Fig. 7j). These findings indicate that while in control conditions

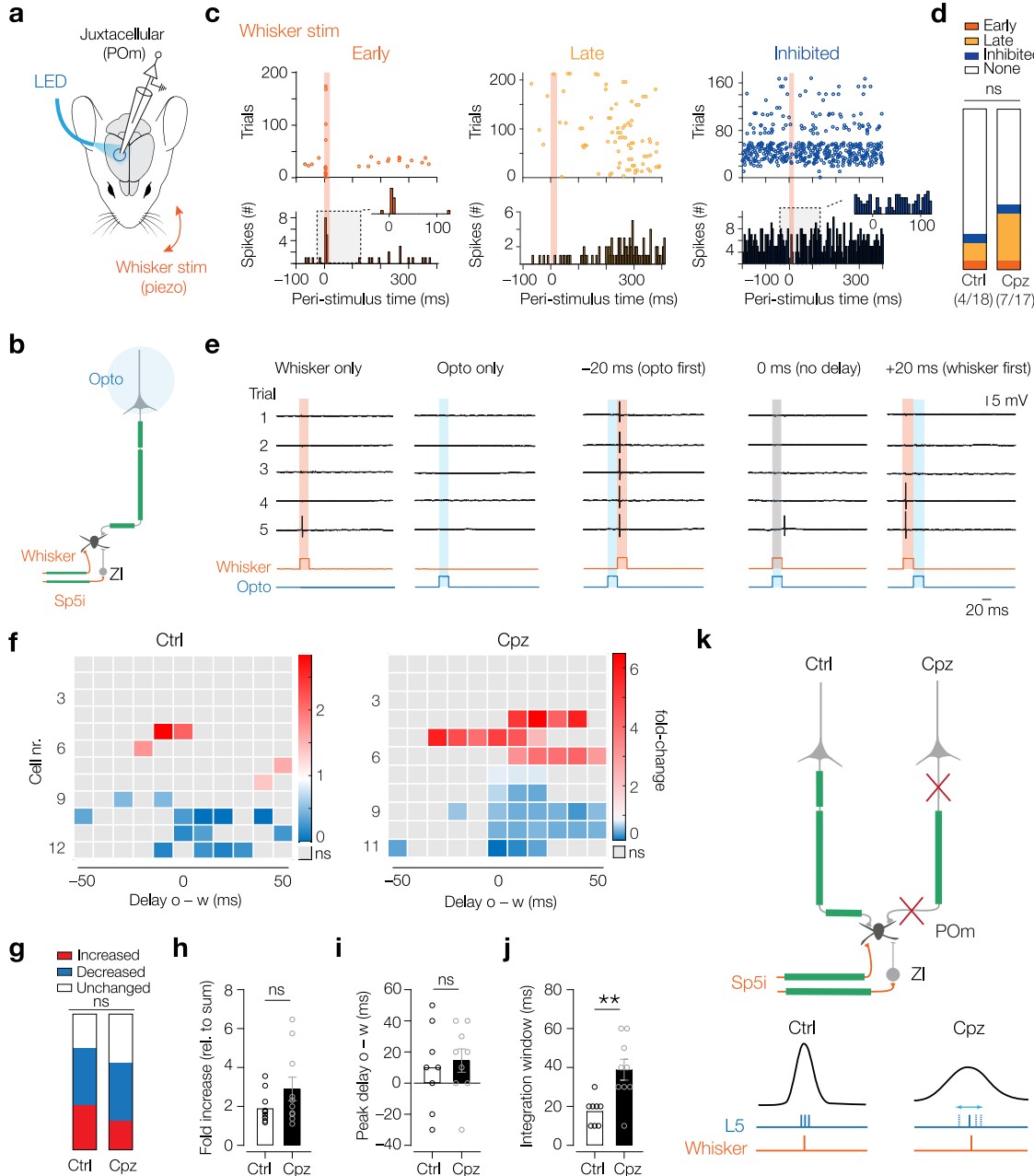

**Fig. 7 | Myelin tunes the temporal integration window for feedback and feed-forward information streams. a** Cortical optogenetic stimulations paired with whisker stimulation during juxtacellular POm recordings (mouse brain adapted from ref. 102). **b** L5 and whisker-evoked inputs via the spinal tri-geminal subnucleus interpolaris (Sp5i) converge on individual POm neurons. Feedforward inhibition via zona incerta (ZI) is generated by whisking. **c** Whisker stimulation (orange shading) produces 4 response types: Early (<50 ms, dark orange), late (>50 ms, light orange), inhibitory (blue, see inset) and no response. **d** Response distribution not significantly different. Fischer's exact test $P = 0.56$. $n = 4$ responsive neurons out of 18 neurons, $N = 12$ mice (Ctrl), $n = 7$ responsive neurons out of 17 neurons, $N = 11$ mice (Cpz). **e** Example responses for opto (blue) and whisker (orange) stimuli generated alone or randomly (between −50 to 50 ms). **f** Heatmaps showing the significantly increased (red) or decreased (blue) spiking responses (opto to whisker [o−w], −50 to +50 ms delay). For each cell, significant changes in spike number was compared to the sum with a Fischer's exact test. **g** Distribution of changes was

not affected (Fischer's exact test $P = 0.88$. $n = 9$ modulated neurons out of 12 neurons, $N = 11$ mice (Ctrl), $n = 9$ modulated neurons out of 14 neurons, $N = 7$ mice (Cpz)). **h** The maximum fold increase of spike numbers was unchanged by demyelination. Mann–Whitney test, $P = 0.29$, $n = 12$ neurons, $N = 10$ mice (Ctrl), $n = 10$ neurons, $N = 7$ mice (Cpz). **i** The delay for largest change (peak) was not significantly affected by demyelination. Unpaired $t$-test $P = 0.46$, $t = 0.75$, $n = 15$ neurons, $N = 11$ mice (Ctrl), $n = 14$ neurons, $N = 8$ mice (Cpz). **j** The window of integration (significantly modulated delays), was significantly greater in Cpz. Unpaired $t$-test **$P = 0.0036$, $t = 3.45$, $n = 8$ neurons, $N = 8$ mice (Ctrl), $n = 9$ neurons, $N = 7$ mice (Cpz). **k** Summary cartoon for the role mye-lination in the L5–POm circuit. Cortical myelin is critical to ensure synchro-nous arrival times of converging inputs as well as coincident input. Partial myelin loss in neocortex produced delays and jittering of timing widening the window for integration. All data are plotted as mean ± SEM. Source data are provided as a Source data file.

POm neurons robustly and precisely encode temporal relationships between sensory inputs and cortical feedback, cortical demyelination reduces the precision thereof. Taken together, in Cpz treated animals, the POm efficiently integrates feedforward driver inputs, reflected by a change in spiking output, but the encoding of timing within the coincidence gate is impeded (Fig. 7k).

## Discussion

In the present study, we found that the unique continuous myelin pattern of long-range projecting L5 pyramidal neurons, starting immediately adjacent to the AIS, plays a critical role in the reliable propagation of the descending burst code and coincident arrival in the thalamus. Activity in the whisker barrel field is linked to the processing of sensory information, which is most notably represented by the canonical operation of L5 pyramidal neuron burst firing[47–49]. The generation of a cluster of 2–4 APs at ~200 Hz, occurs in the primary somatosensory cortex during touch and encodes information of correlated sensory and motor activities[47–52]. At the cellular level, AP bursts are produced by dendritic plateau depolarizations mediating a multiplication of axonal Na⁺ spikes at high frequency in the AIS and proximal axon[47,53,54]. In the absence of sensory input, cortical activity is the main driver of the POm[26,55,56]. In awake rodents, when L5 pyramidal neurons spike at about 3–4 Hz the POm renders in a low gain mode, due to the strong depressing nature of the L5–POm giant synapse[22,27,57]. In this state, the POm acts primarily as a coincidence detector of multiple driver inputs[22], e.g., during neural synchrony of multiple L5 pyramidal neurons firing in a burst mode or coincident activation of brainstem and corticothalamic feedback inputs. Here, we found that demyelination of the L5 pyramidal neuron axons reduced the spiking in the POm evoked by L5 bursts, recorded in vivo in a juxtacellular configuration as well as by Neuropixels (Figs. 3 and 4). The failure to evoke temporally precise POm spike output, even in the face of increased burst generation at the level of the L5 soma, in part can be explained by the increased temporal spread in AP arrival times at giant boutons after demyelination. Since POm neurons typically receive as few as 2–3 inputs from L5 (ref. 27), the asynchrony in synaptic activation may impede the supralinear summation of EPSPs and decrease POm spike probability. The role of neocortical myelin in precisely driving the POm was further identified by the computational model showing that removing the myelin sheath delayed spike arrival times and increased variance of arrival times at the giant bouton (Fig. 6). In addition, simulations of the L5–POm pathway predicted that propagation of spikes >100 Hz may fail along single demyelinated axons and bursts are low pass filtered. This is likely explained by the sustained depolarization of the proximal demyelinated internodes, increasing the time for recovery from inactivation of nodal Na⁺ channels, and impeding the ability to reinitiate the Na⁺ spike within a high-frequency burst. Propagation failure of high-frequency APs traveling along demyelinated L5 pyramidal neuron axons is consistent with experimental in vitro patch-clamp recordings from axons within the neocortex[20]. Although our in vivo recordings and simulations strongly suggest failure of burst propagation, to examine this more directly at the giant boutons, future studies could take advantage of other methods such as in vivo 2-photon imaging. Using 2P imaging from thalamocortical boutons in the cortex a recent study showed that subtle myelination defects impact long-range transmission[9]. Together, the present findings identify a functional role of the dense myelination of L5 in reliably transferring AP bursts from the AIS to postsynaptic targets.

The overall low convergence of L5 input to the POm is thought to be critical to amplify and broadcast the activity of two or three L5 pyramidal neurons to other cortical areas as a *trans*-thalamic pathway. Higher order thalamocortical feedback has been shown to strongly prolong sensory evoked activity in cortical L2/3 and L5 and heighten cortical plasticity[58–61]. This feedback has been proposed to provide contextual information important for perception and cognition[47]. Impaired cortical driving of the POm nucleus after demyelination could therefore have downstream effects on synaptically driven plasticity and sensory processing in the cortex. During sensory activity, in zones of convergent inputs, about 1/3 of POm neurons can act as "AND gates," to detect when L5 feedback co-occurs with ascending input from the brainstem within a narrow time window[24,26,28,61]. Despite the direct excitatory drive from the brainstem, responses to whisker stimulation in POm neurons can be masked by the strong di-synaptic inhibition from the zona incerta under anesthesia[46,62]. In this case, sensory transmission in the POm is contingent upon coinciding cortical activity. Interestingly, consistent with the present observations (Fig. 7), POm neurons display variability in their preferred order and temporal delay of the trigeminal and neocortical inputs[24], raising the possibility that POm is performing temporal computations reflecting their relative timing. Previous work has shown that these computations are crucial for the function of corticothalamic loops. One theoretical model predicts that corticothalamic interactions in the paralemniscal system functions as phase-locked loops. In this model, POm acts as a phase detector that compares the delays between the intended whisking frequency (cortical activity) and the generated whisking frequency (brainstem input)[63,64]. The resulting POm activity is inversely proportional to increased delays, thereby transferring temporally encoded information reported by the whiskers (e.g., on object shape or location) into a rate code[65]. Since demyelination broadened the temporal window of integration and thereby temporally less precise, our findings indicate that the continuous myelination of corticothalamic axons critically shapes the integrative properties in the POm, enabling POm neurons to perform predictive perceptual processing (Figs. 2i and 7k).

In S1 the corticofugal primary axon emerging from L5 becomes myelinated within the first 14 postnatal days, and its development is mediated by both activity- and synaptic vesicle-release independent mechanisms[16,18,66,67]. In contrast, myelination of L2/3 pyramidal neurons starts late in development and oligodendrogenesis and myelin maturation in this region continuous far into adulthood, doubling from young adult to middle aged mice and remaining under influence of ongoing activity-dependent myelination[17,68,69]. To examine the role of the continuous L5 myelin pattern we took advantage of the known heterogeneous effect of cuprizone treatment which differs across dorsoventral and rostro-caudal axis of the brain, but robustly affects the layer 5 and 6 of the neocortex[34,70–72]. Recent time lapse imaging studies during cuprizone-induced demyelination show that oligodendrocyte loss is counteracted by ongoing oligodendrogenesis and remyelination, a process that has been shown to be layer dependent[37,38]. In L5 pyramidal neuron axons a 0.2% 5-week cuprizone treatment produces incomplete myelin loss and even with higher cuprizone concentrations and feeding duration (0.3% and 9 weeks, respectively), patches of myelin can be observed near the white matter[19]. The present study, focusing exclusively on POm-projecting L5 pyramidal neuron axons, corroborated and advanced these findings, by showing that also demyelination of a specific L5 cell type is highly region specific and limited to the axonal regions passing through the neocortex and thalamus (Fig. 5).

At present it is unclear how a toxin which is taken up systemically has layer- and region-dependent effects and profoundly impacts the deeper L5/6 region. Possible explanations could be that oligodendrocyte subtypes have differential sensitivities for mitochondrial respiration impairments induced by the cuprizone toxin[73]. Alternatively, distinct layers or brain regions may contain differentially activated astrocytes and microglia involved in de- and remyelination, or the local neuronal microenvironment for oligodendrogenesis and myelin development varies across brain regions. Interestingly, our layer-specific effects are in line with a recent study utilizing three-

photon imaging to reach the deeper layers and white matter tracts[37]. The authors showed that feeding cuprizone for 3 weeks in the *Mobp-EGFP* mouse line is accompanied by both a reduced oligodendrocyte regeneration and, in addition, the loss of a subpopulation of genetically mature oligodendrocyte subtypes (MOL 5/6) which is not reestablished following cuprizone-mediated demyelination[37]. In future studies, it would be important to identify whether distinct oligodendrocyte subtypes myelinate distinct regions from one neuronal cell type and understand their regenerative potential.

While the present study shows multiple lines of evidence that myelination is critical for the burst propagation and computation of coincidence detection in POm, the specific behavioral role of the L5–POm circuit in sensory processing is not yet fully understood. A large body of literature indicates that cuprizone-treated mice show subtle defects in motor coordination and executive functions but prominent deficits in cognitive tasks. For example, at the peak of demyelination, mice show slower processing speed in tasks involving object recognition, exhibit increased numbers of errors in visuomotor tasks, and more slowly retrieve social memories[72,74,75]. In future research, it should be directly examined how whisker information processing is affected by cuprizone-induced demyelination in awake behaving mice. Studying the whisking behavior in the context of demyelination may help to unravel the functional role of the POm. As opposed to the ventral posteromedial nucleus, the first-order somatosensory thalamic nucleus which is strongly driven by both whisker self-motion and touch[76–79], the role of the POm not yet understood. There is conflicting evidence whether POm neurons are responsive to self-generated whisking, active or passive touch, which likely depends on the stimulation method and behavioral state of the animal[28,56,77,78,80]. Studies in awake, head-fixed mice report that whisker-evoked POm activity mostly encodes coarse movement information, such as mean whisker angle and deflection amplitude, rather than fine phase-locked coding of each whisking cycle and is modulated by arousal[56,79]. Interestingly, cortical silencing increases the acuity of phase representations, indicating that cortical feedback mainly conveys touch signals whereas ascending inputs convey phase information[56,80]. A recent study in head-free mice showed clear tuning of POm neurons to whisker kinematics such as angle and velocity of whisking, which was comparable to VPM, showing that head movements critically shape POm activity, possibly by local integration of motor and sensory information[81]. Our findings that myelin loss causes aberrant integration in the L5–POm corticothalamic circuit could be leveraged to examine the precise role of the POm for processing whisker information and help elucidating the function of temporal encoding of coincident corticothalamic and whisker inputs. With this information, it will be possible to investigate the neuronal underpinnings of POm-specific behavioral paradigms that require the integration of multiple information streams, i.e., sensory signals, generated by whisker deflection by an object, with motor signals generated by whisking, necessary for the detection of horizontal object location[82,83]. Our present findings may lead to exciting new avenues to identify how myelin patterns, and adaptive myelination[10,11,68], are involved in computational processing and mediating complex behaviors.

## Methods
### Animals and ethics
All animal experiments were performed in compliance with the European Communities Council Directive 2010/63/EU effective from 1 January 2013 and with Dutch national law (Wet op de Dierproeven, 1996), reviewed and approved by Central Authority for Scientific Procedures on Animals (CCD, license AVD8010020172426). The specific experimental designs were evaluated and monitored by the Royal Netherlands Academy of Arts and Sciences (KNAW) Animal Ethics Committee (DEC) and Animal Welfare Body (IvD), respectively.

Protocol numbers NIN 20.21.04, NIN 21.21.04. The mouse strain used for this research project, B6.FVB(Cg)-Tg(Rbp4-cre)KL100Gsat/ Mmucd, RRID:MMRRC_037128-UCD, was obtained from the Mutant Mouse Resource and Research Center (MMRRC) at University of California at Davis, an NIH-funded strain repository, and was donated to the MMRRC by MMRRC at University of California, Davis. Made from the original strain (MMRRC:032115) donated by Nathaniel Heintz, PhD, The Rockefeller University, GENSAT and Charles Gerfen, PhD, National Institutes of Health, National Institute of Mental Health. Mice of both sexes were kept at a 12 h light-dark cycle (lights on at 07:00, lights off at 19:00) with ad libitum food and water. Cages were open or IVC cages with corncob bedding, placed within a room with constant ambient temperature of 20–24 °C and a relative humidity of 45–65%. Animals were housed with at least one cage mate. Both control and treatment group were fed ad libitum with the control group receiving powder food for 6 weeks, while the treatment group received a powder food supplemented with 0.2% cuprizone (Cpz, biscyclohexane oxaldihydrazone, Sigma-Aldrich). Fresh powder food was prepared every 2–3 days and bodyweight monitored accordingly. Mice fed with Cpz lost on average $16.8 \pm 4.3\%$ of bodyweight relative to age-matched control mice. Mice with >30% weight loss were considered humane endpoint and excluded from this study (8%, 5 out of 60 animals).

### Viral injections
At 11–12 wks of age, stereotactically guided viral injections were performed under anesthesia with isoflurane (4% induction, 1.5–2% maintenance in oxygen) and analgesia (meloxicam s.c., 5 mg/kg bodyweight prior to surgery, carprofen in drinking water, 0.06 mg/mL for 4 postoperative days). Body temperature was maintained at 37 °C using a heating pad. Eye ointment was applied to prevent eyes from drying out. After drilling of a small craniotomy, 60–80 nl of Cre-dependent expression of humanized ChR2 fused to mCherry (pAAV-EF1a-double-floxed-hChR2(H134R)-mCherry-WPRE-HGHpA) was injected. The virus was a gift from Karl Deisseroth (Addgene plasmid #20297; http://n2t. net/addgene:20297; RRID:Addgene_20297). The virus was injected into the POm (coordinates: 1.7P, 1.25L from bregma, 2.9–3.0 mm depth). For sparse labeling in the case of lightsheet imaging, we used two viruses: One expressing Cre-dependent Flipase (AAV pEF1a-DIO-FLPo-WPRE-hGHpA). The virus was a gift from Li Zhang (Addgene plasmid #87306; http://n2t.net/addgene:87306; RRID:Addgene_87306), diluted 1:20 in S1 L5 neurons (injection coordinates 1.5P, 3.2L, 500–600 μm depth). The other one was a rgAAV expressing Flipase-dependent humanized ChR2 fused to mScarlet (shortCAG-dFRT-hChR2(H134R)_mScarlet, Zürich, plasmid #V522, constructed by the Viral Vector facility, Zürich [hChR2(H134R): p220 and p237; mScarlet-I: Addgene #85065].) injected into POm (undiluted, injection location see above). The virus achieved a sparse and localized targeting of a subset of L5 neurons projecting to the POm enabling visualization and reconstruction of axons projecting to the spinal cord. After ~3 weeks mice were sacrificed for either morphometrical analysis of myelination parameters, acute slice preparation or in vivo recordings.

### Immunohistochemistry
Mice were deeply anaesthetized with pentobarbital (60 mg/kg bodyweight, i.p.) and transcardially perfused with PBS for 5 min, followed by 4% paraformaldehyde (PFA) in PBS. Brains were extracted from the skull and fixated with 4% PFA for 1 h at room temperature (RT). After sucrose cryoprotection steps (1 h 10%, 12 h 20% and 24 h 30%), brains were frozen in sectioning medium (Tissue-Tek®, Sakura). Brains were cut into 40 μm thick thalamocortical slices[84] using a cryostat and were stored in PBS + 0.02% sodium azide until further processing with immunohistochemistry. Slices were blocked with 10% normal goat serum (NGS), 0.5% Triton in PBS for at least 90 min. In the case of

staining for MBP, and slices were blocked for 60 min at 37 °C, followed by 1 h at RT. Subsequently, slices were incubated in primary antibodies (see Supplementary Table 1) diluted in antibody solution (5% NGS, 0.5% Triton), shaking at RT. Incubation times were 1× or 3× overnight (ON) for MBP. Slices were washed for 3 × 5 min in PBS. Afterwards, slices were incubated in secondary antibodies for at least 120 min, diluted in antibody solution. Afterwards, slices were washed for 3 × 5 min in PBS, mounted and covered with mounting medium (Vectashield, Vector laboratories, California, United States).

## Confocal microscopy and image analysis

Imaging was performed using a Leica SP8 X confocal laser-scanning microscope controlled by Leica Application Suite AF (version 3.5.7.23225). Confocal z-stacks of (de)myelinated axons were acquired with a 63× oil-immersion lens (1.4 NA) using the tile-scan function at 10% overlap or the mark and find function and sequential scanning. Step sizes in the z-axis were 0.5–0.6 μm. Images were collected at a 1024 × 1024 or 2048 × 2048 pixel resolution at 100 Hz scan speed, resulting in voxel sizes of ~0.1–0.2 × 0.1–0.2 × 0.5–0.6 μm. Nodes of Ranvier were imaged with a Zeiss Axiovert 200 M controlled by STEDYCON software (Version 9, Abberior Instruments, Göttingen, Germany). Nodes were imaged with a 100× objective (1.46 NA). z-stacks were 4 μm large, step size was 0.25 μm. ROIs were acquired of individual nodes (individual x–y) at a pixel size of 100 nm and a pixel dwell time of 10 μs. Reconstructions of morphometrical parameters were done either in Neurolucida (version 2020.1.3, MBF Bioscience, RRID: SCR_001775) or Imaris Software (Version 9.6.1, Oxford Instruments, Abingdon, England, RRID SCR_007370). Myelin density was measured by reconstructing myelin sheets with the surfaces function in Imaris. Colocalized internodes were reconstructed manually using Neurolucida tracing. Starts and ends of internodes as well as Caspr⁺ puncta and swellings were assigned with individual markers, to measure internode length, percentage of colocalized myelin and swelling size as well as density. Accordingly, AIS length, axon diameter, node-to-node distances (marked by AnkG) were reconstructed in 3D. Putative terminal axonal boutons in POm were outlined manually in 2D at the plane of largest diameter. Caspr density was measured by reconstructing Caspr⁺ puncta in 3D with the spots function in Imaris. Nodal compartments (marked by AnkG and Caspr) of individual nodes of Ranvier were reconstructed with the surfaces function in Imaris. Length and diameter were subsequently measured with the bounding box parameter. The number of mCherry⁺ neurons out of all NeuN⁺ neurons in S1 was counted with ImageJ (FIJI; ImageJ version 1.54f; RRID: SCR_003070).

## Clearing

Clearing of multiple whole brains including the spinal cord using an adjusted version of the iDISCO+ protocol[85]. A mouse, injected with a flipase-dependent rgAAV in the POm and a diluted Cre-dependent flipase into S1 (see section "Viral injections"), was anaesthetized and perfused at the age of 14 weeks. Isolation of the brain with maintaining the spinal cord attached was performed by gently opening the dorsal section of the vertebrae, making an incision in the lower lumbar area of the mouse's spinal column. The mouse's spine was then cut along the entire length. After removing the mouse's skull cap, the whole brain and spinal cord were visible and could be removed by microdissection[86]. Afterwards, the tissue underwent pre-treatment: The brain was fixed in 4% PFA/PBS at 4 °C ON with shaking, then at RT for 1 h. After fixation, the brain was washed in PBS with shaking at RT for 30 min for a total of 3 times. The adjusted iDISCO+ protocol was carried out as follows (all incubation steps while shaking): On the first day, samples were dehydrated in a methanol series, diluted in ddH₂0 (50%, 80%, 100%, 1 h each, 100% for 1 h, RT, shaking). Then, it was bleached in 5% H₂O₂ in 20% DMSO/methanol ON (4 °C). The next day, the sample was washed twice in 100% methanol (30 min at 4 °C,

followed by 3 h at −20 °C), followed by an incubation in 20% DMSO/methanol (RT, 2 h). Afterwards, it was rehydrated by a methanol in ddH₂0 series (80%, 50%, 0%, 1 h each, RT) and washed twice with 1× PBS/0.5% Triton-X-100 (RT, 2 × 1 h). Finally, the sample was incubated in 1× PBS/0.5% Triton-X-100/ 20% DMSO/0.3 M glycine at 37 °C, overnight. Antibody labeling was carried out as follows: Blocking was done in 1× PBS/0.5%, Triton-X-100/10%, DMSO/ 6% and donkey serum at 37 °C for 3 days. Then, the sample was washed in 1× BS/0.2%, Tween-20 with 10 μg/ml heparin (PTwH), at RT (2× for 1 h). The sample was incubated with the primary antibody (anti RFP, see Supplementary Table 1) in PTwH/5%, DMSO/3%, donkey serum, 37 °C for 14 days with 5 mM sodium azide. We refreshed the antibody solution once after 7 days of incubation. Subsequently, it was washed in PTwH, 6× 1 h and 10 min at 37 °C. Incubation with secondary antibody in PTwH / 3% serum (anti gp-Alexa633, see Supplementary Table 1) at 37 °C for 7 days. A second wash with PTwH (6× 1 h 10 min at 37 °C) was performed. For clearing, the sample was dehydrated with a methanol series (20, 40, 60 and 80% 1 h each at RT, 100% 2 × 30 min each). Then, the sample was kept in 66% DiChloroMethane (DCM)/33% methanol ON at RT. The next day there was a 1 h wash in DCM. Finally, the sample was optically cleared with Dibenzyl ether (DBE).

## Lightsheet imaging

The brain and spinal cord were imaged with an Ultramicroscope II (LaVision BioTec) lightsheet microscope equipped with an MVX-10 Zoom Body (Olympus), MVPLAPO 2× Objective lens (Olympus), Neo sCMOS camera (Andor) (2560 × 2160 pixels. Pixel size: 6.5 × 6.5 μm²), and Imspector software (version V380) (LaVision BioTec). Samples were scanned with double-sided illumination, a sheet numerical aperture (NA) of 0.125 (results in a 3.94 μm thick sheet), and a step size of 3.5 μm using the horizontal focusing lightsheet scanning method with 7 steps and using the default algorithm with a sheet width of 40%. The object lens included a standard dipping cap with a working distance of 10 mm. The following laser filter combinations were used: Coherent OBIS 488-50 LX Laser with 545/50 nm filter. A 3 × 6 and a 3 × 9 mosaic scan were conducted with an overlap of 30% for the whole brain and spinal cord, respectively. The scanning of the planes went as follows: 1st z-plane; 2nd x-plane; 3rd y-plane. The brain and spinal cord were sequentially imaged due to image holder constraints. The final voxel size was 1.208179 × 1.208179 × 3.5 μm. The resulting tiff images, of the separate regions, were initially stitched and fused/converted into N5 images using the BigStitcher Plugin in Image (https://imagej. net/plugins/bigstitcher[87]). Next, the spinal cord image was aligned manually to the brain image. Afterwards, the two images were registered with an affine model, regularized with a rigid model, using interest points between the overlapping regions. After the transformation was applied, the images were fused to a single N5 image containing both the brain and spinal cord, with axons sufficiently registered to trace. With the simple neurite tracer (https://imagej.net/plugins/snt/[88]), one axon projecting from S1 to the spinal cord was semi-automatically traced, missing sections in the axon from artefacts due to sample handling were connected manually. Then, a mask of the tracing was created using the fill function. A mesh was extracted from the resulting binary image with the marching cubes function from sci-kit image (https://scikit-image.org/)[89] and exported to.ply with napari-meshio for smoother visualization and recording in 3D using Napari (https://napari.org/stable/, https://zenodo.org/doi/10.5281/zenodo. 3555620). Images were downsampled for visualization but not for tracing.

## Acute slice preparation

At ~14 weeks of age, mice were sacrificed for preparation of acute thalamocortical slices. In brief, mice were deeply anaesthetized by application of pentobarbital i.p. (60 mg/kg), perfused with oxygenated NMDG-ACSF (92 mM NMDG, 30 mM NaHCO₃, 1.25 mM NaH₂PO₄,

2.5 mM KCl, 20 mM HEPES, 25 mM Glucose, 0.5 mM $CaCl_2$, 10 mM $MgCl_2$, saturated with 95% $O_2$ and 5% $CO_2$, pH 7.4 with HCl) and subsequently decapitated. The brain was swiftly removed and submerged in NMDG (composition see above). Three hundred micrometers of thick thalamocortical slices were cut using a Leica VT 1200S vibratome (Leica Biosystems, Wetzlar, Germany). Slices were allowed to recover for <15 min. at 35 °C in NMDG-ACSF. They were subsequently transferred to a holding chamber and kept at RT in holding ACSF (125 mM NaCl, 25 mM $NaHCO_3$, 1.25 mM $NaH_2PO_4$, 3 mM KCl, 25 mM Glucose, 1 mM $CaCl_2$, 6 mM $MgCl_2$, and 1 mM kynurenic acid, saturated with 95% $O_2$ and 5% $CO_2$, pH 7.4).

## In vitro electrophysiology

Slices were transferred to an upright microscope (BX61WI, Olympus Nederland BV) and constantly perfused with oxygenated ACSF: 125 mM NaCl, 25 mM $NaHCO_3$, 1.25 mM $NaH_2PO_4$, 3 mM KCl, 25 mM Glucose, 2 mM $CaCl_2$ and 1 mM $MgCl_2$. The chamber was perfused at a rate of 3 ml/min. Neurons were visualized with a 40× water immersion objective (Achroplan, NA 0.8, IR 40×/0.80 W, Carl Zeiss Microscopy) with infrared optics and oblique contrast illumination. Patch-clamp recordings were performed from morphologically and electrophysiologically confirmed thalamic cells. Patch pipettes were pulled from borosilicate glass (Harvard Apparatus, Edenbridge, Kent, UK) to an open tip resistance of 4–5 MΩ and filled with intracellular solution containing: 130 mM K-Gluconate, 10 mM KCl, 10 mM HEPES, 4 mM Mg-ATP, 0.3 mM $Na_2$-GTP, and 10 mM $Na_2$-phosphocreatine (pH 7.25, ~280 mOsm). Biocytin (3 mg/ml, Sigma-Aldrich) was routinely added to the intracellular solution to allow for post-hoc confirmation of cell morphology and localization. Recordings were performed with a Axopatch 200B (Molecular Devices). Signals were analog low-pass filtered at 10 kHz (Bessel) and digitally sampled at 50 kHz using an A-D converter (ITC-18, HEKA Elektronik Dr. Schulze GmbH, Germany) and the data acquisition software Axograph X (v.1.5.4, Axograph, RRID:SCR_014284, NSW, Australia). Bridge-balance and capacitances were fully compensated in current clamp mode. In voltage-clamp mode the series resistance was compensated to >75%. Input-output curves were generated by injecting 500 ms pulses increasing in 50 pA increments. Single APs were assessed by injecting 3 ms of current pulses at threshold level. Voltage threshold was determined as the voltage at which the first time-derivative exceeded 50 V s$^{-1}$. RMP was measured at $I = 0$. Thalamic neurons receiving corticothalamic driver inputs were identified by flashing brief (5–10 ms) pulses with the 470 nm laser line of a laser diode illuminator (LDI-7, 89 North, USA) through the imaging objective above the cell soma. At first a light-response curve was generated by stimulating with increasing laser intensities (1–15%, corresponding to an output of 1–10 mW, measured with a photodiode). As the curves were not significantly different between treatment groups, all neurons were then stimulated at 10% laser intensity (5 mW). To measure synaptic input strength, somata were excited 10 times for 5 ms at increasing frequencies (1, 10, 20 Hz). Response amplitude was measured in voltage clamp at −73 mV. Recordings were carried out at ~32 °C. A liquid junction potential correction of −13 mV has been applied to all in vitro recordings.

## In vivo recordings

For in vivo recordings, mice were anesthetized with isoflurane (4% induction, 1.2–2% maintenance in oxygen). Breathing rate was constantly monitored and kept around 1–1.5 Hz by adjusting the isoflurane concentration (see also Supplementary Fig. 4). Prior to the start of the surgery, the head was shaved and cleaned of any hair. Mice were placed in the recording setup and the head was fixated with earbars. The mice received a single dose of meloxicam s.c. (5 mg/kg) as well as a topical analgetic (Lidocaine, 10%) on the skin. After placing the incision, the skull was cleaned with ethanol. Craniotomy locations were marked with a micro-ruler relative to Bregma with a needle tip and a

marker (-1.5P, 3.2L for Neuropixels recordings, -1.0P, 3.0L for S1 optic fiber, -1.25L, 1.7P for juxtacellular recordings in POm). Primer (Opti-Bond, Kerr, Switzerland) was applied to the entire skull and cured with UV light. Subsequently, a straight headbar was fixed to the frontal skull with dental cement (Tetric EvoFlow A2, Ivoclar). Next, a recording chamber was modeled with dental cement (Charisma A1, Kulzer, Germany), taking care to leave enough space for the recording probes as well as grounding screws (Neuropixels recordings) or AgCl-pellets. Juxtacellular patch recordings were performed as described in detail in methodological papers[90,91]. Patch pipettes were pulled from borosilicate glass (Harvard Apparatus, Edenbridge, Kent, UK) in a bee stinger shape to an open tip resistance of 5–8 MΩ and filled with extracellular solution: 135 mM NaCl, 5.4 mM KCl, 1.8 mM $CaCl_2$, 1 mM $MgCl_2$, 5 mM HEPES, pH adjusted to 7.2 with NaOH. About 20 mg/ml biocytin was added to the intracellular on the day of recording to enable visualization of pipette tracks as well as successfully filled neurons. Voltage recordings were performed with a Axoclamp 900A (Molecular Devices, San Jose, USA), digitized with a Digidata 1440A (Molecular Devices) and signals acquired with PClamp software (Version 10.7, Molecular Devices). The pipette was advanced with high pressure (>400 mbar) until -100 μm above POm or S1 L5 with a 4-axis micromanipulator (LN Junior, Luigs and Neumann, Ratingen, Germany). Then, the pressure was reduced to 40–50 mbar. Neurons were searched for using 1 μm $z$ steps and identified by a robust increase in pipette resistance (2–3× increase) and extracellular spiking activity >2 mV in current-clamp mode. Simultaneously, an optic fiber (inner diameter 400 μm, Thorlabs, Grünberg, Germany) coupled to an LED driver (LEDD1B, Thorlabs) was placed above S1bf to stimulate L5 neurons expressing ChR2. Responsive neurons in L5 and POm were identified by stimulating with 1 Hz pulses. Only neurons responding within 25 ms and probabilities above 40% at maximum light output power (25 mW at the fiber tip) were included for further analysis. Whiskers were stimulated by placing all whiskers on the contralateral snout into a glass tube, that was connected to a piezo element (PL140.10, Physik Instrumente, Karlsruhe, Germany) controlled by a piezo driver (Physik Instrumente). Both piezo and LED driver were driven by TTL pulses generated by a breakout box connected to a National Instruments module, which was controlled by custom-written MATLAB stimulation protocols via the DAQ function (https://github.com/JorritMontijn/Acquipix).

Recorded traces were pre-processed with Axograph X to detect stimulus and spike times with the event detection function. Delay, variance (calculated as the difference between each point and the mean, squared and divided by $n-1$) and probability of spiking upon a given stimulus were analyzed custom-written scripts in MATLAB. We defined a window relative to each stimulus onset (5–40 ms POm, 2–30 ms L5), during which spikes were classified as evoked, to exclude spontaneous spikes that either were too early or too late to be evoked. Overall, spontaneous spike rates under anesthesia were low (<1 Hz), consistent with previous recordings under anesthesia[27,92]. Thus, the probability of a spontaneous spike occurring during a stimulus was low. We calculated the average delay and variance of the first spike per trial, as well as the overall probability of spiking and average number of spikes during the entire window. If more than one spike occurred during the stimulus, the average number and frequency of burst firing (min 50 Hz) was calculated. Responsiveness of cells to a given stimulus was statistically tested for with the Zenith of Event-based Time-locked Anomalies Test (Zeta-test[93], https://github.com/JorritMontijn/ZETA). After the recordings, mice were perfused with PBS, followed by 4% PFA and postfixed overnight. 100 μm brain slices were cut at a coronal angle, blocked for 1 h (4% NGS, 0.5% Triton), and incubated with streptavidin-Alexa488 conjugate (Thermofischer Scientific) to visualize biocytin. Slices were imaged with a slide-scanner using a 10x objective (Zeiss Axioscan Z1, NA 0.45) and processed with QuPath (Version 0.3.0, https://qupath.github.io/). Pipette tracks as well as

Neuropixels probe tracks were aligned using AP_histology toolbox in MATLAB (https://github.com/petersaj/AP_histology).

## Neuropixels recordings

Neuropixels recordings were performed using a National Instruments I/O PXIe-6341 module and SpikeGLX (https://github.com/billkarsh/SpikeGLX, Version 20201103). Probes were coated in NeuroDiO (Biotium, San Francisco, USA), mounted onto a dovetail holder (Luigs & Neumann, Ratingen, Germany) and inserted into the brain at a 30° angle. The brain was allowed to settle for 10 min before recording was initiated. Then spontaneous activity was recorded for at least 15 min In a subset of recordings we recorded at different anesthesia levels. Here, we recorded for a total of 60 min, and changed the level of anesthesia after 30 min. The first and last 20 min were used for comparison of low and high anesthesia firing rates. During some recordings, heart rate was monitored as an additional parameter of anesthesia depth. Both breathing frequency and heart rate dropped at higher isoflurane concentrations (Supplementary Fig. 5). Raw data was pre-processed with CatGT to remove light-evoked artefacts (https://billkarsh.github.io/SpikeGLX/#catgt). Subsequently, data was spike sorted using Kilosort 2.5 (https://github.com/MouseLand/Kilosort[94]). Good quality single unit clusters were identified by the quality metrics MATLAB pipeline bombcell (https://github.com/Julie-Fabre/bombcell[95]). The correct identification of single units by bombcell was further confirmed in a subset of recordings by manual curation using Phy (https://github.com/cortex-lab/phy). The location of individual units was identified by aligning the probe traces, as extracted with AP_histology (see above), and ephys data with the Universal Probe Finder (https://github.com/JorritMontijn/UniversalProbeFinder[96]). All data, including spike and stimulus times, were compiled for further analysis into a single file with the RecordingProcessor of Acquipix, and data were further analyzed using custom-written MATLAB scripts. Bursting was defined as instantaneous firing frequencies of above 100 Hz. If multiple spikes in a row exceeded this frequency, the entire spike train was considered to be one burst. Neurons needed to fire at least 10 bursts during the entire recording session to be considered a bursting unit. Units with average bursting frequencies > 500 Hz were discarded ($n = 3$ units). To determine whether pairs of units recorded simultaneously in L5 of S1bf and the POm were considered to be putatively connected, we followed a published procedure[97] and computed the spike time cross-correlogram, for all pairs of spikes occurring within 20 ms of each other. We also computed the cross-correlograms of data in which we individually jittered all spike times of both units by a random amount taking from a uniform 20 ms wide distribution, centered at 0 ms. If the height of the peak in the real spike cross-correlogram was at least as high as maximum of the mean jittered spike cross-correlogram plus four times the maximum standard deviation of the jittered spike cross-correlogram, and was at least four spikes high, we considered a pair putatively connected. For the connection probability, we compute the number of putatively connected pairs, divided by the total number of pairs of L5 S1bf and POm neurons that were simultaneously recorded.

## Computational modeling of a L5−POm neuron

A multicompartmental model was created in the NEURON simulation environment (v.7.8.2, RRID:SCR_005393, ref. [98]). The morphology of one L5 pyramidal neuron of a retrograde AAV injected *Rbp4*-Cre mouse with mCherry labeled L5 PNs projecting to POm was selected based on 7 confocal images spanning from the pia in S1bf to POm of and the merged tile scans were imported into the 3D tracing software Neurolucida. We first traced landmarks like cortical areas, white matter, and hippocampus borders in an overview scan at a resolution of 3.7 pixels/μm and a voxel size of $0.27 \times 0.27 \times 1 \, \mu m^3$. Subsequently, we aligned individual regional tile scans (e.g., striatum) to the landmarks and reconstructed multiple long axonal stretches, including nodes of

Ranvier at a resolution of 5.83 pixels/μm and a voxel size of $0.17 \times 0.17 \times 0.6 \, \mu m^3$. The full morphology of the corticothalamic L5 PN axon was obtained by stitching individual axon reconstructions in Neurolucida (Fig. 6a). Anatomical dimensions of the AIS, (juxta)paranode and nodes of Ranvier of the primary axon were included by submicron-precise reconstructions of diameters and length of the immunofluorescence signals for Caspr and AnkG, producing a cylindrical representation of the myelinated L5−POm projection pathway. The 3D Neurolucida data was uploaded into NEURON via the Import3D tool and further specified by creating sectionlists representing 9 different subcellular domains (apical and basal dendrites, the soma and AIS, paranodes, juxtaparanodes, internodes, nodes and axon collaterals).

## Passive properties and myelination

The model simulations were using active and passive properties optimized for selected features of the parameter space to reproduce experimentally measured conduction velocities with biologically realistic myelin sheath properties and saltatory propagation of APs. Myelination was computationally integrated with NEURON's extracellular mechanisms and specified for the internodal, juxtaparanodal, and paranodal sections using optimized double-cable parameters[3]. The myelin sheath was modeled with membrane resistance ($R_{my}$) and capacitance ($C_{my}$) using $R_{ext}$ and $C_{ext}$, respectively, and v and vext[0] represented the neuronal and myelin voltages separated by a submyelin periaxonal resistance ($r_{pa}$) represented by rext[0]. Using rats, L5 pyramidal neuron internode $R_{my}$ and $C_{my}$ have been solved to be 8.56 kΩ cm² and 1.00 μF cm⁻² per myelin membrane, respectively[3]. Pyramidal neuron axons in mice have on average 7−9 lamellae within the corpus callosum[42,43], and 5−8 lamellae in the POm[31]. Hence, to simulate 9 myelin lamellae $R_{my}$ and $C_{my}$ were set to 154.08 kΩ cm² and 0.055 μF cm⁻², respectively. Although myelin thickness may vary between internodes, and the reconstructed axon diameter significantly increased in white matter compared to the gray matter ($1.18 \pm 0.054 \, \mu m$ vs $0.98 \pm 0.052 \, \mu m$, respectively, unpaired *t*-test $P = 0.011$, $n = 39$ WM internodes and 21 internodes in cortex and thalamus) for modeling purposes $R_{my}$ and $C_{my}$ were uniformly assigned to all internodes. Assuming a ~12 nm space between the axolemma and the adaxonal myelin membrane, the specific periaxonal space resistance ($r_{pa}$) was set to 124.9 GΩ cm⁻¹, and paranodal resistance was simulated with an 18.15-fold higher value, as numerically solved previously[3]. The unmyelinated sections of the neuron, including soma and dendrites, the AIS, nodes, and collaterals, were simulated with extracellular mechanisms, but $r_{pa}$ and $r_{pn}$ being infinitely small ($1 \times 10^{-6}$ and $1 \times 10^{-8}$ MΩ cm⁻¹, respectively) and $C_{ext} = 0 \, \mu F \, cm^{-2}$. For all sections, the external voltage (eext) was 0 mV. All neuronal sections contained passive cable properties, including specific cytoplasmic resistivity ($R_i$) of 180 Ω cm and membrane capacitance ($C_m$) of 0.9 μF cm⁻². To simulate the in vivo recorded L5−POm conduction delays the model used the body temperature of 37 °C. To account for the in vivo low input resistance (2−3 fold compared to in vitro recordings) caused by the increased synaptic conductances, higher temperature, increased resting conductance and intact dendrites[45,99,100], we lowered the membrane resistance ($R_m$) 10-fold compared to conventional in vitro models and used a nominal value 2.3 kΩ cm². The final input resistance of the neuron was 13.5 MΩ, approximately 2-fold lower compared to the typical in vitro input resistance of mouse somatosensory L5 pyramidal neurons[19].

## Voltage-gated conductances and synaptic fluctuations

Hodgkin-Huxley (HH) style mathematical differential equation models for voltage-gated ion channels were implemented for fast and persistent Na⁺ channels (Na$_V$ and Na$_P$), K⁺ channels (K$_V$1, K$_V$2/3, and K$_V$7), high- and low-threshold voltage-gated Ca²⁺ channels (CaH and CaT, respectively), a Ca²⁺-activated K⁺ conductance (KCa) model, and for an

HCN channel. In addition, also a cytoplasmic calcium ($Ca^{2+}$) concentration mechanism was also included. The densities were non-uniformly distributed using density values optimized for rat L5 pyramidal neuron models when constraining models to the experimental AP waveform[3,101]. In the absence of detailed in vivo whole-cell recordings of L5 APs, constraining the simulations to experimental data is not possible, and the parameter values described below will not be a unique solution. Conductance density values were adjusted in the AIS and somatodendritic domains to achieve burst firing of 4 APs at 200 Hz when injecting a somatic-current injection (20 ms duration), approximating the optogenetically evoked stimulation (Fig. 6e, c.f. Figs. 1o and 3b). The first AP in the burst showed an amplitude of 100 mV and a halfwidth of 425 μs. In brief, these results were obtained by inserting uniform high densities of $Na_V$ channels but low CaH, CaT, and BK channel densities (30,000, 1.5, 1.5, and 1.0 pS μm$^{-2}$, respectively) in the AIS. $K_V1$ and $K_V7$ conductance densities in the AIS linearly increased to peak densities of 12,000 and 150 pS μm$^{-2}$, respectively. Dendritic sections contained were simulated with distance-dependent linearly decreasing gradients of Na$^+$ channels (from 800 to 25 pS μm$^{-2}$ from soma to the distal dendrites, respectively), as well as a distance-dependent decrease in $K_V1$ and $K_V7$ conductance density. To define the parameters of axonal sections beyond the AIS, we selected parameters to reproduce experimentally realistic forward propagation without failure nor ectopic spike generation. Such a condition was achieved by paranodal sections containing $K_V1$ and $K_V7$ channels at densities of 15 and 1.5 pS μm$^{-2}$, respectively, juxtaparanodes with high $K_V1$ channels (8000 pS μm$^{-2}$), and $K_V7$ channel densities (15 pS μm$^{-2}$) but lacking $Na_V$ channels. Internodal membrane compartments, which are active in the double-cable model, contained modest $Na_V$-, $K_V1$-, and $K_V7$ channel densities (25, 15, and 15 pS μm$^{-2}$, respectively). All nodes of Ranvier contained high densities of $Na_V$ and $K_V7$ channels but low $K_V1$ (30,000, 150, and 15 pS μm$^{-2}$, respectively). Finally, HCN conductance was distributed with a distance-dependent exponential increase in density from soma to distal dendrites and was uniform across all axonal membrane compartments (1 pS μm$^{-2}$). The equilibrium potentials for K$^+$, Na$^+$, HCN, and $Ca^{2+}$ were set to −85, +55, −45, and +140 mV, respectively. The model e_pas was set to −85.0 mV, leading to a −81.8 mV resting potential in the model.

To simulate in vivo-like fluctuations of $V_m$ (Fig. 6e), we used a nonspecific conductance numerically solving excitatory and inhibitory conductances in a Gaussian stochastic manner (https://modeldb. science/8115)[45]. The reversal potentials representing excitatory and inhibitory synapses were set to 0 and −75 mV, with conductance densities of 0.012 ± 0.012 and 0.057 ± 0.0066 μS (mean ± S.D.) and time constants of 2.72 and 10.50 ms, respectively (the "L6 settings" as in Table 1 in ref.[45]). The fluctuating conductance was injected into a section of the primary apical dendrite at 50 μm distance from the soma. Simulations of 150 ms duration were repeated 20 times, with current injections of 1.0 and 1.7 nA (20 ms duration), and the inter-spike intervals were computed by the delay between the peak of the APs in the AIS and a section of a giant presynaptic terminal in the POm.

### Computational simulations of demyelination

Myelin loss was simulated numerically by neutralizing the contributions of the extracellular mechanisms representing $R_{my}$, $C_{my}$, $r_{pa}$, and $r_{pn}$. Specifically, $R_{my}$ was set to $1 \times 10^{-6}$ Ω cm$^2$, $C_{my}$ to 0.9 μF cm$^{-2}$, and $r_{pa}$ and $r_{pn}$ to $1 \times 10^{-6}$ Ω cm$^{-1}$. In the absence of myelin, the peak $Na_V$ conductance densities ($\bar{g}_{Na}$) were distributed to be uniform along the internodal, paranodal, and juxtaparanodal compartments, with a default value of 200 pS μm$^{-2}$, and $K_V1$ and $K_V7$ were uniform at 15 pS μm$^{-2}$. The value of $\bar{g}_{Na}$ was set to the minimum value to enable propagation and below threshold for ectopic APs (Supplementary Fig. 8a). Demyelination with loss of the nodes of Ranvier was simulated by reducing $\bar{g}_{Na}$ in the nodal compartments to internodal densities. All

model simulations were run with 10 μs time steps and the number of segments defined by a d_lambda rule[98] set to 2 kHz. Tests for spatio-temporal accuracy of the simulations by increasing nseg 3-fold showed the CV did not change, indicating a sufficient segmentation.

### Statistics and reproducibility

If not indicated otherwise, data are indicated as mean values and standard error of the mean (SEM). Data were plotted and analyzed in GraphPad Prism 8 software (GraphPad Software, Inc., Version 10.2.3., RRID SCR_002798) and MATLAB (depending on the toolbox or analysis 2018a, 2021b or 2023b, MathWorks, Massachusetts, USA, RRID SCR_001622). For comparisons of two groups data were tested for normality using Shapiro–Wilk test or d'Agostino-Pearson test. Unpaired two-sided $t$-test and Mann–Whitney tests were carried out for parametric and non-parametric comparison, respectively. Wherever possible, datapoints were nested per animal and nested $t$-tests were applied. When comparing two or more groups over multiple time points, we used two-way ANOVA followed by post-hoc correction (details are given in the respective figure legends) for normally distributed data. Kruskal–Wallis test with individually selected multiple comparisons was applied if data were not normally distributed. As all statistics were carried out at 95% confidence intervals, a significant threshold of $P < 0.05$ was used in all analyses. A table summarizing all statistical parameter values has been provided in the Supplementary Data 1.

### Reporting summary

Further information on research design is available in the Nature Portfolio Reporting Summary linked to this article.

## Data availability

All data to evaluate the conclusions are included in the main text and Supplementary Information. Source data are provided with this paper.

## Code availability

Custom-written MATLAB analysis for the in vivo juxtacellular recordings as well as the Neuropixels data can be found at https://github. com/Kolelab/Neuropixels_ephys_Jamann_2025. All NEURON modeling codes are available at https://github.com/Kolelab/long_range-NEURON-model. Openly available code or software from other laboratories was used as follows: ImageJ (FIJI; ImageJ version 1.54f; RRID: SCR_003070), BigStitcher Plugin in ImageJ (Version 2.5.2.), https://imagej.net/plugins/bigstitcher, simple neurite tracer (Version 4.2.1, https://imagej.net/plugins/snt/), sci-kit image (Version 0.22.2, https://scikit-image.org/), Napari (https://napari.org/stable/, https:// zenodo.org/doi/10.5281/zenodo.3555620), Kilosort (v2.5 https:// github.com/MouseLand/Kilosort), bombcell (V1.5.0, https://github. com/Julie-Fabre/bombcell), Phy (V2.0a1, https://github.com/cortex-lab/phy), Universal Probe Finder (V1.1.1, https://github.com/ JorritMontijn/UniversalProbeFinder), Acquipix (V0.9.0, https:// github.com/JorritMontijn/Acquipix), ZETA, AP histology (V2 https:// github.com/petersaj/AP_histology).

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

## Acknowledgements

The authors would like to thank the NIN mechatronics department for their excellent technical support. The authors would especially express their gratitude towards Mike Vink and Andres de Groot for their invaluable contribution to developing the in vivo recording setup. The authors would like to thank Christiaan de Kock (VU Amsterdam) and Alexander Groh (University of Heidelberg) for their valuable feedback on experimental design as well as the manuscript. This work was supported by the Netherlands Research Council NWO Vici 865.17.003 (M.K.) and the KNAW institutional grant (J.M.).

## Author contributions

N.J., M.K.—conceptualization. N.J., J.M., N.P., M.K.—methodology N.J., N.P., D.vd. B., M.B., S.D.—investigation. N.J. J.M., R.L., M.K.—software N.J., J.M., N.P., J.H.—resources N.J., J.M., M.K.—formal analyses. N.J.—data curation. N.J., M.K.—supervision. N.J., M.K.—writing: original draft. N.J., N.P., J.H., R.L., M.K.—writing: review and editing. N.J., M.K.—visualization. N.J., M.K.—project administration. M.K., J.H.—funding acquisition. All authors approved final version of the manuscript.

## Competing interests

The authors declare no competing interests.
