## [Transparent Peer Review file · Nature Communications]

Layer 5 myelination gates corticothalamic coincidence detection

Corresponding Author: Professor Maarten Kole

Version 0:

Reviewer comments:

Reviewer #1

(Remarks to the Author)

Wao, this is a very impressive work. Via empirical tour de force, combining several powerful methods, the authors show that the degree of myelination of cortico-thalamic axons is an important component in the functional tuning of thalamic coincident detection.

Thalamic coincidence detection (CD) has previously been hypothesized to function in two major manners (and possibly more): in detecting strong cortical activations, as part of cortico-thalamo-cortical information transfer, and in detecting cortico-brainstem phase differences, as part of predictive sensory processing (eg, by thalamocortical phase-locked loops). Indeed, as the authors discuss in their paper, there are pieces of evidence supporting both models. Yet, the authors fall short of relating their work to these two models (see below).

Regardless of the incomplete interpretation, the work is of great value. In short, the authors show, in a very elegant manner, that interfering with the level of myelination, converged on through evolution and development, impairs thalamic CD. More specifically, reducing myelination increases the temporal window within which brainstem inputs can still be passed on to the cortex via the P_{Om}. This finding strongly suggests that the specific duration of that temporal window is a parameter that was selected through evolution and development – as the authors rightfully mention, this suggestion has still to be verified behaviorally.

Beyond this important functional insight, the paper contains beautiful data and informative analyses regarding several anatomical and physiological modifications induced by myelin changes, which also provide valuable insights on the basics of corticothalamic processes in general and L5-P_{Om} in particular.

Major comments

1. As mentioned above, the important findings presented in this paper lose some of their power due to lacking interpretation. The authors interpret their results only in relation to the cortico-thalamo-cortical information transfer model, ignoring the predictive cortico-brainstem model. A salient example of the latter is the thalamocortical phase-locked loop (PLL) model, whose descriptions can be found in papers by Ahissar and colleagues along the years. The exact tuning of the temporal width of the thalamic “AND-gate” is crucial in such models, as it determines the efficiency of the predictive processing (see Ahissar 1998, and several of the following works including a 2024 review). Thus, the interpretation here can also have a quantitative aspect.

1. In several places, the description is not clear enough and/or not detailed enough. For example, Figure 3 and associated text – Fig. 3e does not give the impression that “Most of the presynaptic cortical spikes remain subthreshold for spike initiation in the P_{Om}”. Please explain it, and make sure similar ambiguous cases do not occur in other places (see a few examples in the minor comments list). Also, make sure all legends are clear enough (e.g., legend of Fig. 7f – does the figure show changes in delays or changes in responses at various delays?). Please make an effort to verify that everything is clear to all kinds of potential readers, to avoid unnecessarily impairing the quality of the paper.

Minor comments:

1. P. 6, “Interestingly, Ctrl spiking showed two distinct peaks ...” – it is not clear what the authors talk about here – please point to the correct figure.
2. P. 7, last para (“One caveat...”) – the whole paragraph is difficult to understand. See also comment in the Major list. Clarify the ambiguity between the 3:1 reduction and the 90% to 70% reduction.
3. P. 13-14 – “preferred order and delay” – is a bit confusing. I suggest – “preferred temporal relationships” or something alike.

4. P. 14 – “AP firing ... generates sensory percepts ... burst firing” – statement is too strong given available evidence. “during neural synchrony of multiple L5...” – add the possibility of cortico-brainstem synchrony. “Since POM neurons ...” – a typo sentence.
5. P. 15, para 2 (“The overall low...” – add a discussion on thalamocortical NPLs (see Major above). “Typically, whisker stimulation itself rarely...” – see Oram et al, Nat Comm 2024, and fix this statement and related discussion accordingly.
6. P. 17, last para – references to previous works is not correct. Yu et al (Ref 72) showed: 1. VPM conveys both whisking and touch signals (correctly cited by the authors), 2. POM conveys whisking signals (not cited by the authors). Here also, the authors should refer to Oram et al 2024 mentioned above. To broaden their perspective on the thalamocortical system, it is recommended that the authors will look at (not necessarily cite if not needed) Yu et al Cer Cortex 2015 (for S1, S2, VPM and POM complex), and at Ahissar & Oram Cer cx 2015 (for possible functioning).
7. P. 18, “... rather than fine phase-locked coding...” – again, fix this statement according to Oram 2024. “POM-specific behavioral paradigms...” – exactly, this is what is needed. Here you can propose (if you like) tasks requiring the knowledge of whisker location and/or velocity, such as object localization tasks, and compare performance of myelin intact and impaired mice.
8. Fig. 7c – show the early and the inhibited responses at a higher temporal resolution. In the legend fix typo of “>50ms” for early.
9. Fig. 7f – explain better what is shown – the legend and text are not clear and not detailed enough. Also – see in major comments above.

Reviewer #2

(Remarks to the Author)

Decision: Major Revisions

In this manuscript, the authors combined interdisciplinary approaches to examine the impact of cuprizone-induced demyelination on neural signalling between L5 cortical neurons in barrel Ctx and posteromedial nucleus (POM) thalamic neurons. They specifically investigated the impact of cuprizone damage on a set of neuronal features, both in Ctx and Thal, such as onset spike latency, excitability, spontaneous activity and burst coding, using optogenetics and in vivo electrophysiology. Computational modeling revealed that grey matter myelin loss alone could account for a ~5 ms conduction delay, disrupting the burst response and impairing corticothalamic feedback integration/coincidence detection. Their whiskers stimulation suggest that timing disruptions impair coincidence detection in the thalamus, emphasizing the essential role of glia/myelin in ensuring precise neural signaling integrity, and sensory processing.

I think this is an interesting and well-designed study. The manuscript is generally well written. I however have major reservations that prevents me from supporting publication as is, and willing to consider a revised version if the authors decide to do so.

1) Fig 3 suggests a change in excitability caused by cuprizone toxin, in that time to spike onset seems to be advanced with Cpz (e.g. Fig 3d). However the authors state “Together, these results indicate that local synaptic or neuronal changes could not explain the differences in demyelination-induced spike probability.” Previous studies have indicated changes in neuronal excitability following demyelination (<https://doi.org/10.1523/JNEUROSCI.4747-14.2015>), even in the thalamocortical circuit (see <https://doi.org/10.1186/s12974-016-0629-0>). It would be important for the authors to contextualize these rather unexpected findings.

2) The computational results are, as written, rather difficult to judge. This is problematic as the conclusions derived from modelling are central to the claims made by the authors.

I. I was unable to determine how the authors “optimized” their code, as claimed? The methods indicate that they mainly used standard parameter values available in the literature, as its usually done in such multi-compartmental software environment such as NEURON. But how did the author implement/model cuprizone damage? As far as I know, this is a very complex computational problem, that involves selecting which and how many compartments will suffer damage, how they are spatially organized, how this damage is actually implemented, and how/if ion channel/active conductances are involved (if at all). Details provided in the Methods are not sufficient to help any reader understand how this was done, and the caption of Fig 6 only states “MBP, AnkG and Caspr signals guided the assignment of myelinated (green) sections”.

II. How many neuron morphologies were tested? The use of singular in both the methods and Fig 6 caption suggests only 1 neuron was considered, but I doubt its the case. Please elaborate.

III. Fig6f suggests that random voltage fluctuations were involved in the model, which explain the variance in spike timing. How was the variance of this noise chosen/optimized? Is it the same across all compartments? Further, if conduction velocities were computed using voltage time series, then these measurements are also variable and hence should be examined across independent repeated realizations/trials (both dynamics, and morphology, if more than one are used). However the conduction velocities reported in the manuscript were not mentioned with any error margin.

3) Am I correct in saying that the authors didn’t measure or characterize any of the predicted burst propagation failures in their experiments (aside from simulations)? I didn’t manage to see support for the the burst code failure in fig 3, except from the title of the figure.

4) Linked to the point above, wouldnt the (expected) change in spiking probability (presumably linked to changes in excitability?) reported in fig 3 be consistent with the manifest depolarization seen in fig 3h between ctrl (-54mV) and cpz

(-42mV)? Its unclear whether the author did measure those or simply fail to report these quantities.

5) The thalamocortical system is known to be involved in the emergence of up/down, slow oscillations closely linked to burst-like neural activity (<https://doi.org/10.1016/j.conb.2014.10.003>), in which time delay (and hence the integrity of myelinated fibers) is believed to play a critical role. However, it is my understanding that the authors did not see any difference in burst like behavior (Fig 4) in the thalamus. I was left a bit puzzled by the results of fig 4, and left wondering 1) how to reconcile these with the claims that cpz impairs the burst code and 2) how these claims would hold if one was to recontextualize the results of Fig 4 using a time-sensitive (but rate invariant) metric, such as ISI for instance? While the mean firing quantities measured might not differ with respect to control, their variabilities seem to be impacted, especially if cuprizone-induced spike time jitter is involved.

6) A recent paper by Borden et al. ([https://www.cell.com/neuron/fulltext/S0896-6273\(22\)00546-3?dgcid=raven_jbs_etoc_email](https://www.cell.com/neuron/fulltext/S0896-6273(22)00546-3?dgcid=raven_jbs_etoc_email)) showed that timing and synchronization, rather than response magnitude, are central to thalamocortical sensory processing, in mice. They used optogenetics and cortical voltage imaging to show that thalamic hyperpolarization amplifies sensory-evoked thalamic bursting, but not cortical responses. It would be important for the authors to motivate and compare their results to those of Borden et al, to provide some baseline for the effects they are trying to characterize.

Minor:

I do not understand "...negative (o before w) to positive (w before o) delays.."

"...probability. nor the delay to..."

"are suggesting burst failures to examine this more directly"

Use CV for either coefficient of variation or conduction velocity

Version 1:

Reviewer comments:

Reviewer #1

(Remarks to the Author)

Thank you for the thoughtful revision -i have no concerns left.
congratulations for this wonderful contribution to science.

(Remarks on code availability)

Reviewer #2

(Remarks to the Author)

The authors have addressed all my concerns.

(Remarks on code availability)

We thank the reviewers for their positive evaluation and constructive input. Below, we have provided a detailed point by point response to the comments and concerns with the reviewers' text in black italic, our responses in blue and text cited from the manuscript in red italics.

Reviewer #1

Wao, this is a very impressive work. Via empirical tour de force, combining several powerful methods, the authors show that the degree of myelination of cortico-thalamic axons is an important component in the functional tuning of thalamic coincident detection. Thalamic coincidence detection (CD) has previously been hypothesized to function in two major manners (and possibly more): in detecting strong cortical activations, as part of cortico-thalamo-cortical information transfer, and in detecting cortico-brainstem phase differences, as part of predictive sensory processing (eg, by thalamocortical phase-locked loops). Indeed, as the authors discuss in their paper, there are pieces of evidence supporting both models. Yet, the authors fall short of relating their work to these two models (see below).

Regardless of the incomplete interpretation, the work is of great value. In short, the authors show, in a very elegant manner, that interfering with the level of myelination, converged on through evolution and development, impairs thalamic CD. More specifically, reducing myelination increases the temporal window within which brainstem inputs can still be passed on to the cortex via the POm. This finding strongly suggests that the specific duration of that temporal window is a parameter that was selected through evolution and development – as the authors rightfully mention, this suggestion has still to be verified behaviorally. Beyond this important functional insight, the paper contains beautiful data and informative analyses regarding several anatomical and physiological modifications induced by myelin changes, which also provide valuable insights on the basics of corticothalamic processes in general and L5-POm in particular.

We thank the reviewer for their positive remarks and glad they recognize the valuable new insights into the role of myelin in corticothalamic processing. We fully agree that the work could be better positioned within the existing models of corticothalamic information processing and have included them in our discussion.

Major comments

1. As mentioned above, the important findings presented in this paper lose some of their power due to lacking interpretation. The authors interpret their results only in relation to the cortico-thalamo-cortical information transfer model, ignoring the predictive cortico-brainstem model. A salient example of the latter is the thalamocortical phase-locked loop (PLL) model, whose descriptions can be found in papers by Ahissar and colleagues along the years. The exact tuning of the temporal width of the thalamic “AND-gate” is crucial in such models, as it determines the efficiency of the predictive processing (see Ahissar 1998, and several of the following works including a 2024 review). Thus, the interpretation here can also have a quantitative aspect.

The reviewer provides highly valuable context which we missed to include in the background literature and interpretation in the first submission. We have now included the PLL model and accompanying literature in the discussion of our results (p.16).

2. In several places, the description is not clear enough and/or not detailed enough. For example, Figure 3 and associated text – Fig. 3e does not give the impression that “Most of the presynaptic cortical spikes remain subthreshold for spike initiation in the P_{Om}”. Please explain it, and make sure similar ambiguous cases do not occur in other places (see a few examples in the minor comments list).

The original text was indeed unclear. We have revised the manuscript text and made sure to remove the ambiguity throughout. For Fig. 3e we changed the text to (at p.7):

“One caveat in the analysis of spike transmission probability is the strong frequency-dependent synaptic depression of the giant L5–P_{Om} synapse, typically translating L5 bursts of 3–4 APs from multiple neurons into 1–2 APs in the P_{Om}^{22,24,27}. In other words, while the first L5 spike in the burst gets reliably transmitted at high probability, most of the subsequent depolarizations at the giant terminal will not translate into P_{Om} spikes, despite a reliable propagation from the AIS to the giant terminal of all presynaptic spikes within a burst. Due to this filtering, detection of transmission failure along the demyelinated axon based on somatic recordings alone is challenging. Even when some presynaptic spikes fail, the remaining ones could still drive the P_{Om} to spike (Fig. 3b,d). To obtain a more detailed insight into putative failure of spike transfer within the burst code, we made additional, temporal analyses to detect which spikes within the presynaptic cortical burst are reliably transmitted. We overlaid all spikes recorded in P_{Om} neurons for a given light intensity (25 mW) and temporally aligned these to the optogenetically evoked first and second spike in the presynaptic burst (Fig. 3e). The analysis revealed a significant reduction of P_{Om} spike probability for the spike in the first cluster (from ~90 to ~70%, Fig. 3f), indicating an increased failure for the first spike in a burst.”

Also, make sure all legends are clear enough (e.g., legend of Fig. 7f – does the figure show changes in delays or changes in responses at various delays?). Please make an effort to verify that everything is clear to all kinds of potential readers, to avoid unnecessarily impairing the quality of the paper.

We thank the reviewer for indicating the incomplete legends. In the revised version we have made sure they are complete.

Minor comments:

1. P. 6, “Interestingly, Ctrl spiking showed two distinct peaks ...” – it is not clear what the authors talk about here – please point to the correct figure.

We included the correct figure reference (Fig. 2i) and to improve clarity we plotted the control and cuprizone spike histograms separately. The lack of a first peak is now well visible.

2. P. 7, last para (“One caveat...” – the whole paragraph is difficult to understand. See also comment in the Major list. Clarify the ambiguity between the 3:1 reduction and the 90% to 70% reduction.

Thank you for pointing out the unclarity. In the revised version we have provided a more detailed explanation of the reason for our analysis to analyze the P_{Om} spiking in temporally aligned clusters of spikes. The text has been rewritten into a more detailed explanation (see above).

3. P. 13-14 – “preferred order and delay” – is a bit confusing. I suggest – “preferred temporal relationships” or something alike.

Thank you for the suggestion. We have rewritten it to *“These findings indicate that while in control conditions P_{Om} neurons robustly and precisely encode temporal relationships*

between sensory inputs and cortical feedback, cortical demyelination reduces the precision thereof” and made the final statement of this paragraph more concise (p. 14).

4. P. 14 – “AP firing ... generates sensory percepts ... burst firing” – statement is too strong given available evidence.

Thank you for the suggestion. We have rephrased the sentence to “*Activity in the whisker barrel field is linked to the processing of sensory information.*”

“during neural synchrony of multiple L5...” – add the possibility of cortico-brainstem synchrony. “Since P_{Om} neurons ...” – a typo sentence.

We have rephrased the sentence to “*during neural synchrony of multiple L5 pyramidal neurons firing in a burst mode or coincident activation of brainstem and corticothalamic feedback inputs*” and corrected the typo.

5. P. 15, para 2 (“The overall low...” – add a discussion on thalamocortical PLLs (see Major above). “Typically, whisker stimulation itself rarely...” – see Oram et al, Nat Comm 2024, and fix this statement and related discussion accordingly.

The findings by Oram et al Nat Comm 2024 are interesting to include, and in the revision we added the thalamocortical PLLs. We have rephrased the sentence on the possible role of the inhibitory inputs in gating corticothalamic feedback to be less misleading on whisking induced P_{Om} activity. However, we decided to include the suggested findings by Oram et al on whisking induced P_{Om} activity in the later paragraphs on P_{Om} encoding of whisker information, since we believe it was more fitting in this section, enriching the more in-depth discussion of the role of P_{Om} encoding (p.18-19).

6. P. 17, last para – references to previous works is not correct. Yu et al (Ref 72) showed: 1. VPM conveys both whisking and touch signals (correctly cited by the authors), 2. P_{Om} conveys whisking signals (not cited by the authors). Here also, the authors should refer to Oram et al 2024 mentioned above. To broaden their perspective on the thalamocortical system, it is recommended that the authors will look at (not necessarily cite if not needed) Yu et al Cer Cortex 2015 (for S1, S2, VPM and P_{Om} complex), and at Ahissar & Oram Cer cx 2015 (for possible functioning).

Thank you for pointing out the findings of Oram et al 2024, showing representation of whisker kinematics in P_{Om} when mice are allowed to roam head-free, as well as the review by Yu et al Cer Cortex 2015. These studies were helpful to provide context in the Discussion.

7. P. 18, “... rather than fine phase-locked coding...” – again, fix this statement according to Oram 2024. “P_{Om}-specific behavioral paradigms...” – exactly, this is what is needed. Here you can propose (if you like) tasks requiring the knowledge of whisker location and/or velocity, such as object localization tasks, and compare performance of myelin intact and impaired mice.

As stated above, we have included the findings of Oram et al. 2024. In addition, we have included the suggested proposed experiments for P_{Om} specific paradigms at p. 18–19.

8. Fig. 7c – show the early and the inhibited responses at a higher temporal resolution. In the legend fix typo of “>50ms” for early.

Thank you for the suggestion. We fixed the typo and have also changed the figure and figure legend to show the data at higher temporal resolution using insets.

9. Fig. 7f – explain better what is shown – the legend and text are not clear and not detailed enough. Also – see in major comments above.

We have included a more detailed explanation of the figure in the text and legend to clarify what is depicted.

Reviewer #2

In this manuscript, the authors combined interdisciplinary approaches to examine the impact of cuprizone-induced demyelination on neural signalling between L5 cortical neurons in barrel Ctx and posteromedial nucleus (POm) thalamic neurons. They specifically investigated the impact of cuprizone damage on a set of neuronal features, both in Ctx and Thal, such as onset spike latency, excitability, spontaneous activity and burst coding, using optogenetics and in vivo electrophysiology. Computational modeling revealed that grey matter myelin loss alone could account for a ~5 ms conduction delay, disrupting the burst response and impairing corticothalamic feedback integration/coincidence detection. Their whiskers stimulation suggest that timing disruptions impair coincidence detection in the thalamus, emphasizing the essential role of glia/myelin in ensuring precise neural signaling integrity, and sensory processing. I think this is an interesting and well-designed study. The manuscript is generally well written. I however have major reservations that prevents me from supporting publication as is, and willing to consider a revised version if the authors decide to do so.

We thank the reviewer for the constructive evaluation of our manuscript. In response to the concerns, we have obtained new data on the spatiotemporal relationships of neuronal excitability in L5 and POm and revised the manuscript. The changes are clarified in our point-by-point responses below and we believe this has been very helpful in improving the manuscript. We also carefully and thoroughly revised the manuscript for minor issues.

1) Fig 3 suggests a change in excitability caused by cuprizone toxin, in that time to spike onset seems to be advanced with Cpz (e.g. Fig 3d). However, the authors state “Together, these results indicate that local synaptic or neuronal changes could not explain the differences in demyelination-induced spike probability.” Previous studies have indicated changes in neuronal excitability following demyelination (<https://doi.org/10.1523/JNEUROSCI.4747-14.2015>), even in the thalamocortical circuit (see <https://doi.org/10.1186/s12974-016-0629-0>). It would be important for the authors to contextualize these rather unexpected findings.

We agree our data indicate an earlier onset of optogenetically-evoked burst in L5. In fact, this was mentioned in the original submission and analyzed in Fig. 2f. At p.6: “**Importantly, this power-delay relationship was only slightly affected by demyelination, with Cpz firing significantly earlier at low light power without change in variance (Fig. 2f).**” We have not further investigated the mechanism to the temporally advanced onset but there is substantial evidence, including from our own preceding and present research, that cuprizone-induced demyelination increases neuronal excitability, by increasing excitation and reducing inhibition. In vitro whole-cell patch-clamp recordings as well as Neuropixels and longitudinal in vivo EEG recordings show that Cpz treatment increases pyramidal

neuron burst firing and reduces PV interneuron excitability and synaptic inhibition in S1, and ultimately at the network level causes interictal epileptic-like discharges during specific behavioral states (Hamada and Kole, 2015; Bacmeister et al, 2020, Dubey et al., 2021).

In the revised version of the manuscript, we have further analyzed the Neuropixels dataset and quantified the units in L5 and P0m to compute spike frequency, fraction of bursting across all spikes. The new results plotted in Fig. 4 showed that, in line with our acute slice recordings (Supplementary Fig. 4), the excitability in P0m is not changed. In contrast, in L5 AP bursts occurred significantly more often and within the burst exhibited higher spike frequencies. Importantly, however, the increased spontaneous L5 pyramidal neuron excitability cannot account for the impaired and delayed evoked L5–P0m transmission. We realized that the concluding sentence at p. 8 was perhaps unclear. We have rewritten it to read: *“Together, these results indicate that neither the L5 synaptic properties nor the neuronal changes can explain the observed impaired L5–P0m spike transfer probability induced by demyelination.”*

2) *The computational results are, as written, rather difficult to judge. This is problematic as the conclusions derived from modelling are central to the claims made by the authors. I. I was unable to determine how the authors “optimized” their code, as claimed? The methods indicate that they mainly used standard parameter values available in the literature, as its usually done in such multi-compartmental software environment such as NEURON.*

We thank the reviewer for asking further clarification. In the revised manuscript we have extensively rewritten the methods for the simulations and explained how we selected the parameter settings (see Methods, computational modeling, p. 28–30). In brief, the starting point for including HH conductance models and passive parameter values was based on previously optimized models from mouse and rat layer 5 pyramidal neurons (see Hamada and Kole, 2015; Cohen et al., 2020). In those studies, many of the model parameters were obtained by using NEURONs multirun fitting algorithms constrained to the voltage responses under passive as well as active conditions from neurons recorded in acute slices. Admittedly, the word ‘optimization’ in the context of the model is perhaps inappropriate; to the best of our knowledge there are no in vivo patch-clamp recordings from mouse L5 pyramidal neurons which can be used to constrain and optimize the model parameters at sub milliseconds resolution. Instead, we simulated in vivo-like conditions by lowering input resistance, insert fluctuating inputs and made sure the control model neuron reproduced a burst of 4 APs at ~200 Hz propagating with a CV of approximately 1.9 m/s. At p. 29: *“The model simulations were using active and passive properties optimized for selected features of the parameter space to reproduce experimentally measured conduction velocities with biologically realistic myelin sheath properties and saltatory propagation of APs.”*

But how did the author implement/model cuprizone damage? As far as I know, this is a very complex computational problem, that involves selecting which and how many compartments will suffer damage, how they are spatially organized, how this damage is actually implemented, and how/if ion channel/active conductances are involved (if at all). Details provided in the Methods are not sufficient to help any reader understand how this was done, and the caption of Fig 6 only states “MBP, AnkG and Caspr signals guided the assignment of myelinated (green) sections”.

We agree demyelination is complex with both time- and region-dependent variation of the nodal domains. Note that the characterization of the L5–P0m axons after cuprizone

treatment in this study revealed at least 8 types of nodal structures (Fig. 5, Supplementary Fig. 6) and simulating all the possible morphologies is not tractable. We simulated demyelination for three scenarios: 1) by only removing the myelin layers, 2) removing myelin as well as the nodal domains and 3) adding axonal swellings. In the revised manuscript we made the approach clearer by rewriting the Methods section called *Computational simulations of demyelination* (p. 31) and by creating new Supplementary Figs. 7 and 8.

II. How many neuron morphologies were tested? The use of singular in both the methods and Fig 6 caption suggests only 1 neuron was considered, but I doubt its the case. Please elaborate.

We created one morphologically realistic model in which internodes and nodes were included based on fluorescent markers. As described previously in the Methods (p. 28), multiple axonal regions from the same imaging plane were used to make one composite morphology, producing 4.2 mm from L5 to the P_{Om}, an abstraction of the L5–P_{Om} neurons. To better show the morphology we created a new Supplementary Fig. 7.

III. Fig6f suggests that random voltage fluctuations were involved in the model, which explain the variance in spike timing. How was the variance of this noise chosen/optimized? Is it the same across all compartments?

Indeed, we used noise-like fluctuations to examine the role of myelin in burst propagation (Fig. 6e-g). We used the point-conductance model as developed by Destexhe et al. (Ref. 45 in the manuscript) which is based on in vivo sharp electrode recordings from cat layer 5 pyramidal neurons and biophysically detailed models. The nonspecific conductance fluctuations are numerically solved by an Ornstein-Uhlenbeck stochastic process. We used the same settings for reversals and standard deviations as published (see p. 31). While it is possible to distribute synaptic fluctuations across all dendritic sections this becomes computationally demanding, making run times orders of magnitude longer and unnecessary. We aimed to test how the AP travels along the partially demyelinated regions of the axon when firing thresholds are irregular due to in-vivo like V_m fluctuations.

Further, if conduction velocities were computed using voltage time series, then these measurements are also variable and hence should be examined across independent repeated realizations/trials (both dynamics, and morphology, if more than one are used). However the conduction velocities reported in the manuscript were not mentioned with any error margin.

In the revised figure we now report not only CV values from the deterministic models but also for the stochastic conditions, now plotted in Fig. 6f.

3) Am I correct in saying that the authors didn't measure or characterize any of the predicted burst propagation failures in their experiments (aside from simulations)? I didn't manage to see support for the burst code failure in fig 3, except from the title of the figure.

This is not entirely correct. In the original we showed that with optogenetically evoked L5 bursts demyelination caused a reduced probability of P_{Om} spikes within the first milliseconds (Fig. 3f). In the revised manuscript we have added new analyses on the Neuropixels recording (Fig. 4). We discovered that within paired units of L5 and P_{Om}, putatively synaptically connected, demyelination generated less spikes within the first 10 milliseconds temporally correlated with L5 bursts. Together with the model simulations we conclude that myelin is required to enable temporally synchronized spikes in the P_{Om}.

4) Linked to the point above, wouldn't the (expected) change in spiking probability (presumably linked to changes in excitability?) reported in fig 3 be consistent with the manifest depolarization seen in fig 3h between ctrl (-54mV) and cpz (-42mV)? Its unclear whether the author did measure those or simply fail to report these quantities.

The resting membrane potentials we recorded whole cell in vivo were from a low number of neurons (n = 3), prohibiting drawing a meaningful conclusion about the population data. We removed these values from the revised Fig. 3. However, we collected multiple lines of evidence that the POM neurons were not affected in excitability following cuprizone-induced demyelination. First, in vitro slice recordings from POM neither showed resting membrane potentials or intrinsic property changes (Supplemental Fig. 4). Secondly, our new Neuropixels analysis confirms the similarity in excitability; there were no differences in POM spike frequency, fraction of burst events, fraction of bursting units and burst frequency (Fig. 4). Hence, the impaired evoked L5–POM responses more likely represents failure to transfer the high frequency burst.

5) The thalamocortical system is known to be involved in the emergence of up/down, slow oscillations closely linked to burst-like neural activity (<https://doi.org/10.1016/j.conb.2014.10.003>), in which time delay (and hence the integrity of myelinated fibers) is believed to play a critical role. However, it is my understanding that the authors did not see any difference in burst like behavior (Fig 4) in the thalamus.

This is correct, despite the fact there is some demyelination locally in the POM, the POM intrinsic excitability and synaptic drive is unchanged (see point 4 above).

I was left a bit puzzled by the results of fig 4, and left wondering 1) how to reconcile these with the claims that cpz impairs the burst code and 2) how these claims would hold if one was to recontextualize the results of Fig 4 using a time-sensitive (but rate invariant) metric, such as ISI for instance? While the mean firing quantities measured might not differ with respect to control, their variabilities seem to be impacted, especially if cuprizone-induced spike time jitter is involved.

We have rigorously updated the analyses for Fig. 4 and now extended the work by 1) including analyses for spontaneous burst firing and 2) identifying the putatively connected neuronal pairs within the Neuropixels probe recordings. In line with juxtacellular recordings these data are supporting the idea that demyelination causes L5 bursts failing to temporally cluster spikes in the POM.

6) A recent paper by Borden et al. ([https://www.cell.com/neuron/fulltext/S0896-6273\(22\)00546-3](https://www.cell.com/neuron/fulltext/S0896-6273(22)00546-3)) showed that timing and synchronization, rather than response magnitude, are central to thalamocortical sensory processing, in mice. They used optogenetics and cortical voltage imaging to show that thalamic hyperpolarization amplifies sensory-evoked thalamic bursting, but not cortical responses. It would be important for the authors to motivate and compare their results to those of Borden et al, to provide some baseline for the effects they are trying to characterize.

We thank the reviewer for notifying us on this interesting study. The study highlights the role of hyperpolarization in the VPM for thalamic bursting and subsequent cortical responses. As we did not measure VPM responses, and neither did we observe changes in the probability to produce POM bursts with hyperpolarizing current steps, or changes to POM neuronal

intrinsic excitability (see above), we are unsure how our results can be compared to the findings of Borden et al. and have therefore chosen to focus the discussion on the role of P0m coincidence detection in sensory perception..

Minor:

I do not understand "...negative (o before w) to positive (w before o) delays..",

We removed the acronyms and now write more accessible "opto before whisker". Defined in the legend since we use O and W in the figure.

"...probability. nor the delay to..."

Revised. The point should have been a comma. Now reads: "... with increased probability, nor the delay to the peak of the integration (**Fig. 7h, i**)."

"are suggesting burst failures to examine this more directly"

Changed wording into "are suggesting burst failure of propagation ... "

Use CV for either coefficient of variation or conduction velocity

In the revision we have now consistently used 'variance' to describe the spread of values and CV when referring to 'conduction velocity'.